# An Improved Empirical Model for Flood Discharge Atomization and Its Application to Optimize the Flip Bucket of the Nazixia Project

**DOI:** 10.3390/ijerph16030316

**Published:** 2019-01-24

**Authors:** Jijian Lian, Junling He, Fang Liu, Danjie Ran, Xiaoqun Wang, Chang Wang

**Affiliations:** State Key Laboratory of Hydraulic Engineering Simulation and Safety, Tianjin University, 92 Weijin Road, Nankai District, Tianjin 300072, China; jjlian@tju.edu.cn (J.L.); 1017205048@tju.edu.cn (J.H.); 1018205051@tju.edu.cn (D.R.); 1014205052@tju.edu.cn (X.W.); jgwangchang@tju.edu.cn (C.W.)

**Keywords:** atomization, flip bucket, nappe wind, stochastic splash model, trajectory nappe, two-phase flow

## Abstract

Flood discharge atomization is a serious challenge that threatens the daily lives of the residents around the dam area as well as the safety of the water conservancy project. This research aims to improve the prediction accuracy of the stochastic splash model. A physical model test with four types of flip bucket is conducted to obtain the hydraulic parameters of the impinging outer edge of the water jet, the relationship of the splashing droplet diameter with its corresponding velocity, and the spatial distribution of the downstream nappe wind. The factors mentioned above are introduced to formulate the empirical model. The rule obtained from the numerical analyses is compared with the results of the physical model test and the prototype observations, which yields a solid agreement. The numerical results indicate that the powerhouse is no longer in the heavy rain area when adopting the flip bucket whose curved surface is attached to the left wall. The rainfall intensity of the powerhouse is significantly weaker than that of other types under the designed condition, so we choose it as the recommended bucket type. Meanwhile, we compare the rainfall intensity distribution of the original bucket and the recommended bucket under different discharge which rates ranging from 150.71 to 1094.9 m^3^/s. It is found that the powerhouse and the owner camp are no longer in the heavy rain area under all of the working conditions. Finally, it is shown that the atomization influence during the flood discharge can be reduced by using the recommended bucket.

## 1. Introduction

A significant number of high dam projects have been built in deep and narrow valleys. When the water is discharged from the high dam reservoir into the plunge pool, the great energy contained by water can threaten the stability of the plunge pool [1,2,3]. To ensure the safety of the plunge pool, researchers have proposed many bucket shapes to reduce the unit energy of the water jet, such as the slit-type bucket, the dentated bucket, the diffusion bucket, the oblique bucket, the tongue-shaped bucket, and so on [4,5,6]. However, the atomization of the flow nappe discharged from a high dam is intensified when the unit energy is reduced [7], and rainstorms caused by flood discharge atomization are much heavier than natural rainfall [8,9], which can threaten the operation of the hydropower station, the safety of traffic, the stability of the downstream slope as well as influencing the surrounding ecological environment. Besides, the atomization rainfall and diffuse clouds during the flood discharge increase air humidity, dampen houses, and introduce various diseases to residents. In particular, for the silt-carrying flow, flood atomization is often accompanied by mud fog, and the small diameter of the mud fog attached to the leaves can cause the obstruction of stomata, thus blocking photosynthesis and damaging the environment [10,11,12]. Therefore, designing a reasonable layout and bucket type for hydro projects and predicting the intensity distribution of atomization rainfall are particularly important.

There are three method types used in present atomization research. Prototype observation is an important method to study atomization, but this type of measurement is difficult, and the observation results are easily affected by random factors. In addition, it is hard to directly extend the observation results of one specific project to another. The physical model test is an extension and supplementation to prototype observation, and as it is not subject to time and environment constraints, repetitive tests and quantitative descriptions can be carried out. However, the physical model test is usually designed according to the Froude similitude, and it would bring great uncertainty to the atomization model similarity, which is greatly affected by surface tension and air buoyancy [13]. Many researchers have studied the scale effects of the physical model. Wu et al. [12] proposed that when the Weber number is greater than 500, the influences of surface tension and viscous force are relatively weak and have little effect on the test results. While there is a close relationship between the Weber number and the model’s geometric dimensions, the similarity relations between the model and the prototype results should be explored further [13]. Therefore, the research on flood discharge atomization is still dominated by numerical calculations [14]. Numerical calculations are a semi-empirical method based on prototype observation and the physical model test. Liu et al. [15,16,17,18] studied the characteristics of splashing droplets and established a mathematical model to analyze the aeration characteristics of jet-flow, the movement of splashing droplets and the progress of the jet impacts on the downstream water surface to estimate the splash area and the total splash flow. Zhang et al. [19,20] and Lian et al. [21] established a stochastic model of the flood discharge atomization and proposed the three-dimensional rain-fog transportation method, and the Runge-Kutta and Monte-Carlo theories were used to work out the corresponding differential equation. Based on the research of Lian and Zhang, Liu et al. [14] developed the stochastic mathematical model by studying the atomization mechanism of jet-flow. They profoundly analyzed the influences of the various parameters of the model on the rainfall intensity distribution, and their results were verified in the Lijiaxia, Wujiang and Ertan hydropower stations. Liu et al. [22] also improved the stochastic splash model on the basis of its predecessors and verified the accuracy of the mathematical model by an indoor splash test and field measurement. In previous research, the intelligent algorithm has been widely used in hydraulic engineering [23,24,25,26,27,28], for example, Dai et al. [29] and Peng et al. [30] studied the flooding atomization mechanism and used the the back-propagation neural network model to simulate the rainfall intensity distribution of the Laxiwa and Manwan Hydropower Station. Liu et al. [31] proposed a mixture neural network model based on the radial basis function to quantitatively predict the rainfall intensity distribution of Dongjiang Hydropower Station.

However, the previous mathematical models have some disadvantages, as follows: ① the empirical formula has poor accuracy in calculating the hydraulic characteristics (the pitch, width, height, and velocity of flow) of the tongue-shaped bucket, the curved surface bucket, and other extraordinary-shaped buckets, which would cause large errors in calculation of the impinging width of the water jet; ② previous research did not give a detailed description of the spatial distribution of nappe wind but instead used the generalized function; and ③ the correlation between the diameters and the corresponding velocities of the splashing droplet is neglected. While the factors mentioned above have great influences on rainfall intensity and distribution, to improve the prediction accuracy of the mathematical splash model, a physical model test is conducted to obtain the characteristics of the hydraulic parameters of the impinging outer edge of the water jet, the dimensionless relationship between the droplet diameter and the splashing velocity, and the spatial distribution of the downstream nappe wind. The improved model is verified by prototype observation and the physical model test, which yields a solid agreement. Meanwhile the rainfall intensity distribution of different buckets is calculated by the improved model, and the recommended flip bucket is obtained. This paper provides a reference for the established water conservancy project which has caused huge economic losses without fully considering the atomization problem. 

## 2. The Physical Model and Mathematical Model of Flood Discharge Atomization

### 2.1. The Physical Model

#### 2.1.1. Layout and Measurement of the Physical Model Test

The hydraulic model of the Nazixia Hydropower Station starting from the diversion canal to 0 + 11 m of the downstream channel, which includes the dam, spillway, flood discharging tunnel, and downstream channel, and the spillway consists of an inlet channel section, lock chamber section, draining section, and spout section. The model’s geometric scale is 1:50, and it was designed according to Froude similitude. The system’s circulating water is provided by the water pump. The supplement tank has a length of 10 m, a width of 12 m, and a height of 6 m; the downstream channel has a length of 9.5 m, a width of 4.2 m, and a height of 0.2 m. The height between the flip bucket and downstream water surface is 0.33 m. The topography of the reservoir area and the downstream channel was determined by the actual topography. The surface of the terrain is coated with cement mortar, and the outlet structures are made of plexiglass. The test system is shown in Figure 1. The spillway prefers to release flood water, and the maximum discharge flow and outlet velocity are 1141 m^3^/s and 35.82 m/s, respectively.

The flood discharge was measured by a rectangular weir. Through the dragging force between the trajectory nappe and the surrounding air, the surrounding air generates nappe wind by obtaining additional momentum from the high-speed flow nappe. The nappe wind was measured by the ultrasonic wind sensor (maximal measuring range: 0~10 m/s; measuring accuracy: 0.01 m/s; equipped with acquisition system; each point measuring 400 s, acquisition frequency: 1 Hz). The droplet diameter and splashing velocity were measured by the laser raindrop spectrograph (diameter measurement range: 0.125–8.5 mm; velocity measurement range: 0.2–20 m/s; intensity measurement range: 0.005–250 mm/h) which can collect and classify a large number of the characteristics of droplet parameters based on the droplet diameter. The flow velocity was measured by the propeller’s current meter which has a measuring accuracy of 0.01 m/s. We designed a splash collecting device and arranged it on the left side slope; the size of the device is shown in Figure 1. The sponge collection boxes with lengths of 20 cm and widths of 10 cm were numbered and arranged at the bottom of the left slope near the powerhouse tailrace; number 1 was located at point B and the interval of the measuring point was 20 cm. The atomization condition of each bucket type was compared by measuring the weight of the splash per unit time (the maximum range of the electronic scale is 4 kg, and the accuracy is 0.1 g). Starting from the downstream far boundary of the splashing area, the sponge boxes were placed point by point from far to near, and the number and initial time of each test box was recorded at the same time. After a period of time, the collection boxes were collected from strong to weak splashing intensity one by one. Riverbed scouring and silting were measured by a leveling instrument, and the digital camera was used to record the scouring of the terrain. The similarity method of the scouring velocity is often used to simulate the scour of rock foundation; according to the local rock geology, the scouring velocity of rock foundation is determined. Based on the scouring velocity data supported by the Geological Survey Institute, we used the Isbash formula [32] ($v=K\sqrt{D}$, *v* represents the scouring velocity (m/s); D represents the sediment grain size (m), *K* represents the lithology coefficient and its range is from 5 to 7) to calculate the diameter range of the prototype, and then we obtained the diameter range of the physical model.

#### 2.1.2. Flip Bucket Types and Operating Conditions

Generally, there are three types of flip bucket for trajectory energy dissipation: ① the constant width-type; ② the expansion-type; and ③ the contraction-type. In this research, we mainly focused on the established water conservancy project, and we optimized the shape of the bucket with the principles of minimum construction and best effect; therefore, we selected a typical flip bucket of each type to study their atomization characteristics, as shown in the Figure 2. Shape 1 is the original bucket of the hydropower station. At the same time, we found that the downstream scouring and silting of different buckets increased with the bucket angle decreased, and when the bucket angle was too small, this could seriously reduce the output of the power station. The splashing droplets and nappe wind near the powerhouse obviously increased with an increase in the bucket angle. When the bucket angle is too big, this can seriously aggravate the rainfall intensity. Therefore, we chose a bucket angle of 50° to study the rainfall intensity distribution of different flip buckets, and the test conditions are shown in Table 1.

### 2.2. Basic Theory of the Improved Mathematical Model

#### 2.2.1. The Number of Splashed Water Droplets

When the jet flow impinges with the downstream water surface, taking the continuous bucket for example, a rectangle with a length dζ and width dξ is selected around any point on the outer boundary of the jet. The water volume *q* through the rectangle in unit time can be calculated as [13]:
(1)$$q=\frac{1}{2}\kappa Cv_{z}hl$$
where κ=2dξ/h, based on the feedback analysis of the prototype and experiment data, κ=0.005∼0.03, and the coefficient varies for different buckets. For the continuous bucket, the value of κ ranges from 0.005–0.01, for the silt bucket and twisted bucket, the value of κ ranges from 0.01–0.03. *h* is the thickness of the jet flow, which is obtained by the physical model test, $v_{z}$ is the projection of water droplet velocity in the Z-axis. Provided that *q* is completely converted to the splashed water droplets, the number (*n*) of splashed water droplets per unit time can be expressed as $n=\frac{6q}{\pi d_{50}^{3}}$. This has been verified in our previous research [13]; therefore, we adopted it in this research. $d_{50}$ is the average diameter of water droplets, and a previous study [13] showed that the value of $d_{50}$ is usually selected as a constant that equals 3 mm. *C* and *C_M_* are the water concentration coefficients at any point, and the maximum water concentration coefficient in the same cross-section, respectively. The relationship between *C* and *C_M_* can be expressed as [13,33]:(2)$$\frac{C}{C_{M}}=e^{-\pi \left[{\left(2\xi /h\right)}^{2}+{\left(2\zeta /b\right)}^{2}\right]}$$
(3)$$C_{M}=\frac{\bar{C}}{0.494}$$
where C¯=α1Fr−53; *C_M_* and $\bar{C}$ are the maximum and average water concentration coefficients in the same cross-section, respectively. ξ and ζ are the distances from the measurement point to the jet centers in two curvilinear directions, *b* is the width of the jet flow, Fr is the Froude number, which can be expressed as $Fr=v/\sqrt{gh_{w}}$, $h_{w}$ represents the thickness of the jet flow without aeration, $\alpha_{1}$ can be expressed as $\alpha_{1}=3.80Fr_{0}-4.75$, and $Fr_{0}$ is the initial Froude number on the outlet of the bucket.

The impinging width of water jet *l* has a great influence on the rainfall intensity distribution. The previous mathematical model of flood discharge atomization calculated the trajectory distance and the width of the impinging outer edge by the empirical formula [13,14,20,21]. However, *l* varies from one bucket to another. For special shaped buckets (the tongue-shaped bucket and the twisted bucket), the flow regime is extremely complicated, and it is difficult to calculate the width and impact location of water jet impingement accurately with the empirical formula. So, we used the physical model test to obtain the specific hydraulic parameters of the impinging outer edge of the water jet which forms a linear ejection source, and we divided it into multiple segments.

#### 2.2.2. Stochastic Model of Splash Water Droplets 

Due to the large velocity of the trajectory nappe, the outer edge of the jet is severely aerated and discontinuous and it cannot completely enter the downstream water surface [15], and most droplets splash around, which provides the main source of atomization rainfall [34]. The mechanism of flood discharge atomization is extremely complex, and it is tremendously difficult to use the theoretical derivation method to obtain the mechanism. So, we conducted many splashing tests and gathered prototype data to obtain the semi-empirical formula, and it has been verified to be applicable for predicting the atomization by many researchers [13,14,20,21]. In addition, Duan et al. [34] and Liu et al. [35] compared the calculated results of the stochastic splash model with the experimental data, and their results indicated that the gamma distribution assumption of the diameter and initial velocity can simulate the atomization phenomena very well; therefore, we adopted this approach in this research. The randomness of splashing water droplets is described as the following:
(1)The diameter d of a water droplet obeys a Gamma distribution [13]:(4)$$f\left(d\right)=\frac{1}{{b_{1}}^{a_{1}}\Gamma \left(a_{1}\right)}d^{a_{1}-1}e^{-\frac{d}{b_{1}}}$$
where $a_{1}=2$, $b_{1}=0.5d_{50}$, $d_{50}$ is the mode of the droplet’s diameter.(2)The initial velocity $v_{0}$ of a water droplet obeys a Gamma distribution [13]:(5)$$f\left(v_{0}\right)=\frac{1}{{b_{2}}^{a_{2}}\Gamma \left(a_{2}\right)}{v_{0}}^{a_{2}-1}e^{-\frac{v_{0}}{b_{2}}},$$
where $a_{2}=0.25\bar{v_{0}}$, $b_{2}=4$, $\bar{v_{0}}$ is the average velocity of the water droplet.

In previous research, the velocity and diameter of water droplets were taken as independent variables [13,14,20,21,22]. However, we found that there is a certain correlation between the splashing droplet diameter and its velocity in the experiment. As a droplet’s diameter increases, the ejection requires a larger initial impulse. So, for droplets with large diameters, the splash velocity and splash area are relatively small, and the relationship between velocity and diameter has a great influence on simulating the rainfall intensity distribution. Therefore, we used a laser raindrop spectrograph to obtain the dimensionless relationship between the droplet diameter and splashing velocity, as shown in Figure 3. Through data fitting, the dimensionless empirical equation can be expressed as the following:(6)$$\frac{\bar{v_{0}}}{{\bar{v_{0}}}_{ave}}=11.25\cdot \exp(-\frac{d/d_{ave}}{0.38});\;R^{2}=0.98,$$
where $\bar{v_{0}}$ represents the average velocity of the water droplet for each diameter. d is the diameter of a splashing droplet, ${\bar{v_{0}}}_{ave}$ and $d_{ave}$ represent their average respective values.
(3)The initial elevation angle β of the water droplet obeys the Gamma distribution [13].
(7)$$f\left(\beta \right)=\frac{1}{{b_{3}}^{a_{3}}\Gamma \left(a\right)}\beta^{a_{3}-1}e^{-\frac{\beta }{b_{3}}},$$
where $a_{3}=10\beta_{mo}+1$, $b_{3}=0.1$, $\beta_{mo}$ is the mode of the initial elevation angle of water droplet.(4)The initial azimuth angle φ of the water droplet obeys a normal distribution [13].
(8)$$f\left(\varphi \right)=\frac{1}{\sigma \sqrt{2\pi }}e^{-\frac{{\left(\varphi -\mu \right)}^{2}}{2\sigma^{2}}},$$
where $\mu =0^{\circ }\sim 5^{\circ }$, $\sigma =20^{\circ }\sim 30^{\circ }$.

#### 2.2.3. Nappe Wind of Flood Discharge

Since there is a violent interaction between water and air, the rapid splashing of water droplets and water masses will bring a nappe wind within a certain range. Additionally, the nappe wind will accelerate the water droplets and water masses to spread further, which has a great influence on the rainfall intensity and coverage areas of the flood discharge atomization. Due to the mechanism of nappe wind being extremely complex, there is little research on the distribution of nappe wind, and most previous mathematical models of flood discharge use a generalized function to determine it. However, the spatial distribution of nappe wind greatly differs from one bucket to another, so we used the ultrasonic wind sensor to determine the detailed spatial distribution of the nappe wind. 

The dimensionless relationship between the nappe wind and the longitudinal distance is shown in Figure 4, We defined point B, which is the intersection of the spillway axis and power station tailrace, as the starting point, which is shown in the Figure 1. The results prove that the nappe wind tends to attenuate with a decrease in the distance to point B. In the spillway axial direction, the nappe wind of shape 1 is smaller than that of the other shapes. The main reason for this is that the nappe wind is attenuated by the action of viscous forces and frictional forces, as the bucket angle of shape 1 is bigger than that of the others, which causes the jet trajectory length to be shorter than that of the others, and the distance between the measuring point and the impinging position of the jet as well as the effecting time of the viscous forces and frictional forces are longer than those of the others. Therefore, the nappe wind of shape 1 is the smallest among the four types. In addition, due to being affected by the shape of the bucket, the attenuation degree of nappe wind is different for each bucket type. Shape 1 and shape 2 are tongue-shaped buckets, and their nappe wind attenuates slowly along the axis. Shape 3 and shape 4 are a curved surface bucket and a continuous bucket, respectively. The attenuation degree of nappe wind in the axis direction is larger than that of tongue-shaped bucket. At point B, Figure 5 is the dimensionless relationship between the nappe wind and the vertical distance. We can see that in the vertical direction, the nappe wind attenuation degree of shape 3 in the axis direction is larger than that of the other buckets.

At the same time, the dimensionless relationship between the nappe wind and the cross-distance is obtained, and as shown in Figure 6, the cross-distribution of the nappe wind varies significantly from one bucket to another. As shape 4 is the continuous bucket, the nappe wind satisfies the normal distribution, and the peak value of the nappe wind appears in the axis of the spillway. With a longer distance between the measuring point and the spillway axis, the nappe wind decreases more rapidly. Shape 1 and shape 2 are tongue-shaped buckets. As shown in Figure 6, we found that the peak value of their nappe winds appeared in the both sides of the spillway axis, which is obviously different from the continuous bucket. Additionally, shape 1 is an asymmetric tongue-shaped bucket, and the nappe wind on both sides of the spillway axis is obviously different. The main reason for this is that the outward expansion of the left side wall increases the proportion of water jet on the left side, which makes the nappe wind of the left side obviously larger than that of the right side. For shape 3, as an effect of the curved surface attached to the left wall, the water jet turns to the right side of the spillway axis as does the nappe wind, which causes a rapid decrease in the nappe wind from left to right; this is obviously different from other buckets. Therefore, the previous mathematical model of flood discharge atomization which the nappe wind obeys the normal distribution function is no longer applicable. Previous research indicated that the maximum nappe wind velocity near the impinging position is approximately equal to one-third of the velocity of the flow nappe [36]. So, we can obtain the downstream spatial distribution of the prototypical nappe wind.

#### 2.2.4. The Motion Equations of the Water Droplets

Based on the random model of splash water droplets, a group of pseudo random numbers for the diameter, initial velocity, initial elevation angle, and azimuth angle are generated by using the Monte-Carlo theory [13,19,20,21], which forms the splash water droplets. The splashed water droplets are restricted by gravity, nappe wind, air resistance, and buoyancy. Therefore, the motion equation of each water droplet can be expressed as [13,14,22]
(9)$$x_{i}^{\prime \prime }=-\frac{3}{4}\frac{C_{f}}{d}\frac{\rho_{a}}{\rho_{w}}\left|\vec{v}-\vec{u}\right|\left(v_{i}-u_{i}\right)+\left(\frac{\rho_{a}}{\rho_{w}}-1\right)g_{i}$$
where *i* = 1, 2, 3 represent the longitudinal, lateral, and vertical directions, respectively, $x_{i}^{\prime \prime }$ is the acceleration component of the water droplet, $v_{fi}$ is velocity component of the wind, $v_{i}$ is the velocity component of the water droplet, $g_{i}$ is the gravity acceleration component. The initial conditions are $x_{i}(0)=0$, $x_{1}^{\prime }(0)=v_{0}\cos\beta \cos\phi$, $x_{2}^{\prime }(0)=v_{0}\cos\beta \sin\phi$, $x_{3}^{\prime }(0)=v_{0}\sin\beta$. *ρ*_a_ and *ρ*_w_ represent the air density and water density, respectively. The Runge–Kutta method [13,19,20,21] is used to calculate the motion differential equations; therefore, the rainfall intensity distribution can be obtained. *C_f_* is the resistance coefficient; its value is relative to the Reynolds number Re=(u−vf)d/ν, and it can be expressed as [34]
(10)Cf={24Re; Re≤0.224Re(1+0.15Re0.687); 0.2≤Re≤8000.44; Re≥800.

## 3. Verification of the Improved Mathematical Model 

### 3.1. Verification by Prototype Observation 

The rainfall area of flood discharge atomization is divided into three levels [14]: the torrential rainstorm area (the rainfall intensity is more than 50 mm/h), the rainstorm area (the rainfall intensity ranges from 10 to 50 mm/h), the drizzling rain area (the rainfall intensity ranges from 0.5 to 10 mm/h). We conducted prototype observation of the rainfall intensity of flood discharge atomization at Nazixia Hydropower Station on 4 July 2017. During the flood discharge period, the discharge of the spillway was 418.5 m^3^/s which is less than half of the designed discharge. The field data of the rainfall intensity supported by the administration of Nazixia hydropower station indicated that the powerhouse is partly in the torrential rainstorm area, and a photograph of the powerhouse rainfall is shown in Figure 7. The numerical results are shown in the Figure 8, the powerhouse is still in the rainstorm area and partly in the torrential rainstorm area. Therefore, the numerical results correspond well to the prototype observations.

### 3.2. Verification by the Experimental Results

#### 3.2.1. Verification of Different Flip Buckets the Designed Condition

(1) Numerical result of the rainfall intensity distribution

The improved mathematical model was used to calculate the rainfall intensity distribution of different buckets under the designed condition. As shown in Figure 9, the powerhouse was in the rainstorm area when adopting shape 1 (60°), shape 2 (50°), and shape 4 (50°), so the rainfall intensity around the powerhouse was not reduced by adjusting the angle of the flip bucket, the lateral diffusion degree of water jet, or the width of the impinging edge. However, for the shape 3, due to the effect of the curved surface attached to the left wall, the water jet and the peak value of the nappe wind turned away from the powerhouse, and the powerhouse and owner camp were no longer in the rainstorm area. This shows that the shape 3 is the best type to reduce the rainfall intensity around the powerhouse among the four bucket types.

(2) Experimental results

The rainfall intensity distribution around the powerhouse is mainly subject to the splashing water weight and the nappe wind on the upstream slope of the powerhouse tailrace. Thus, we used them as the control indicators to study the rainfall intensity distribution.


***The splashing water weight***


In this paper, a splash collecting device was designed to collect the splashing droplets on the upstream slope of the powerhouse tailrace under the designed condition. The splash collecting device and the measuring point arrangement are shown in Figure 1. The total weight of the splashing droplets is shown in Table 2. Compared with shape 1, the splash weights of the shape 2, shape 3, and shape 4 buckets decreased greatly, and the splash weights of the shape 2, shape 3, and shape 4 only accounted for 69.58%, 0.69%, and 7.25% of that of shape 1, respectively. The main reason for this is that shape 1 and shape 2 are tongue-shaped buckets, and compared with shape 3 (skew bucket) and shape 4 (continuous bucket), the lateral diffusion of their aerated jet is more intensive, and the width of the impinging edge is relatively wide, which results in the range of splashing droplets being broader, so the splash weights of shape 1 and shape 2 are obviously larger. For shape 3 and shape 4, as an effect of the bucket type, the lateral diffusion degree of the aerated jet and the width of the impinging edge are reduced, especially for shape 3, which is affected by the curved surface that is attached to the left wall. The water jet and the impact point turn to the right side of the spillway axis, which gives shape 3 an advantage in reducing the splash weights.

Under the action of nappe wind, the atomization rainfall around the powerhouse is greatly affected by the splash weight at the bottom of the left slope near the powerhouse tailrace. The splash weights of different bucket types were measured by the sponge collection boxes under the designed condition. As shown in Figure 10, for the weight of the splash water from measuring points 1 to 10, the splash weight has a decreasing trend. The splash weight of shape 2 is generally larger than that of other buckets. This is mainly because shape 1 and shape 2 are tongue-shaped buckets which makes the width of the impinging edge and the range of splashing droplets obviously larger than those of shape 3 and shape 4, so the splash weights of shape 1 and shape 2 are obviously larger than those of other buckets. Comparing shape 1 with shape 2, the bucket angle of shape 2 is smaller than that of shape 1, which causes the trajectory length of shape 2 to be relatively larger, and the distance between the measuring point and the impact point of jet has a great influence on the splash weight. Therefore, the splash weight of shape 2 is significantly larger than that of shape 1. Shape 3 and shape 4 have the same bucket angle but are affected by the action of the skew bucket. The splash weight of shape 3 is obviously smaller than that of shape 4. Therefore, shape 3 is more advantageous for reducing the weight of the splash water at the bottom of the left slope near the powerhouse tailrace.


***The nappe wind on the upstream left slope of the tailrace***


The nappe wind on the upstream left slope of the tailrace has a great influence on the rainfall intensity around the powerhouse, so we used the ultrasonic wind sensor to measure the nappe wind. We set 10 control points, numbered 1–10, to measure the nappe wind on the upstream left slope of the tailrace. We regarded point A as the first point, and the interval of the measuring length was 44 cm. The locations of the other points are shown in Figure 1. The distribution of nappe wind on the left slope is shown in Figure 11, the nappe wind of all the buckets tended to decrease from point 1 to point 10, and the nappe wind of the shape 1 attenuated much slower than the others. The main reason for this is that the asymmetry tongue-type and its larger bucket angle lead to the height of the trajectory nappe and the width of the impinging outer edge being relatively larger, which makes the trajectory nappe have a wide range of action, so the nappe wind of shape 1 attenuates much more slowly than the others. The nappe wind of shape 3 and shape 4 attenuate rapidly, which is mainly because the trajectory nappe of shape 3 and shape 4 are more contracted, and the impinging outer edges of shape 3 and shape 4 are obviously smaller than those of shape 1 and shape 2, which makes the action range of the trajectory nappe relatively small, so the nappe wind of shape 3 and shape 4 attenuate more rapidly than those of shape 1 and shape 2. The nappe wind around the powerhouse of the shape 3 is much less than that of the shape 4. In addition, since the scour hole of shape 4 is very deep so the sedimentation of the downstream river bed blocks the tailrace outlet completely, which impacts on the power generation seriously. We found that the shape 3 works best to reduce the nappe wind.

From the above analysis, it was determined that both the splash weight and the nappe wind of shape 3 on the left slope near the powerhouse are significantly smaller than that of the others. Therefore, shape 3 is regarded as the recommended bucket; the experimental results agree well with the numerical results.

#### 3.2.2. Verification of the Original and Recommended Buckets Under Different Discharge Conditions

(1) Numerical result for the rainfall intensity distribution

To compare the rainfall intensity distribution of the recommended bucket with that of the original bucket under different working conditions, the optimized mathematical model was used to calculate the rainfall intensity distribution, and the numerical results are shown in Figure 12. The intensity and coverage area of the rainfall reduced as the discharge decreased. The rainfall coverage area of the recommended bucket shifted to the right side of the spillway axis. This is mainly because the impact point and the peak value of the nappe wind turned to the right side of the spillway axis. The rainfall intensity around the powerhouse is significantly reduced when we adopted the recommended bucket, and the powerhouse was no longer in the rainstorm area under different working conditions.

(2) Experimental results 


***The splashing water weight***


The total splash weights of the original and recommended buckets under seven conditions are shown in Table 3. The splash weight of the recommended bucket was much lighter than that of the original bucket, and the splash weight of the recommended bucket only accounted for 3.79% of the original splash weight under the check condition. The main reason for this is that the lateral diffusion degree of the aerated jet is reduced and the trajectory nappe turns to the right side of the spillway axis, far away from the powerhouse. In addition, the bucket angle of the recommend bucket is smaller than that of the original bucket, which reduces the height of the trajectory nappe so the heights of the splash droplets falling into the splash collecting device are also significantly reduced. Therefore, the splash weight of the recommended bucket is much lighter than that of the original bucket under all of the discharge conditions. 

The splash weights of the original bucket and recommended bucket at the bottom of the left slope under seven conditions are shown in Figure 13. The splash weight of the recommended bucket was obviously smaller than that of the original bucket under all of the working conditions. The lateral diffusion of the aerated jet and the width of the impinging edge reduced as the flood discharge decreased. The splash weight of both the original bucket and the recommended bucket decreased as the distance between the measuring point and the powerhouse increased. However, for condition 5 to condition 7, lighter splash droplets could be collected because of the small discharge flow and large distance between the impact location and the powerhouse. Therefore, the recommended bucket has an obvious advantage in reducing the rainfall intensity near the powerhouse under all of the working conditions.


***Nappe wind on the upstream slope of the powerhouse tailrace***


The nappe wind of the original bucket and the recommended bucket on the upstream slope of the powerhouse under seven conditions are shown in Figure 14. The nappe wind of the recommended bucket was also obviously smaller than that of the original bucket. The recommended bucket obviously suppressed the nappe wind on the left slope near the powerhouse. We found that the nappe wind of the original bucket attenuated slowly. This is mainly because the height and lateral diffusion of the trajectory nappe is large, which makes it act widely. However, for the recommended bucket, the width of the impinging outer edge and lateral diffusion of the trajectory nappe reduced which was affected by the bucket type. 

The splash weight and nappe wind of the original bucket were compared with those of the recommended bucket on the left slope of the powerhouse tailrace under seven conditions. The splash weight and nappe wind of the recommended bucket were obviously smaller than those of the original bucket. Therefore, the recommended bucket has a remarkable effect on improving flood discharge atomization around the powerhouse, which is consistent with the numerical simulation results.

## 4. Conclusions

In this research, we presented an improved mathematical model for flood discharge atomization. In the process of model improvement, the hydraulic parameters of the impinging outer edge of water jet and the spatial distribution of the downstream nappe wind, especially the relationship between the splashing droplet diameter and its velocity obtained from the physical model test, were considered. The results from the improved mathematical model were evaluated by prototype observation and the physical model test, and they showed satisfactory agreement, so the improved mathematical model is reasonable.

We used the improved mathematical model to calculate the rainfall intensity distribution of different buckets. It was found that the rainfall intensity around the powerhouse and the owner camp was significantly weaker than that of the other bucket types when we adopted shape 3, so shape 3 is the recommended bucket. From comparing the rainfall intensity distribution of the recommended bucket with that of the original bucket, we found that both the intensity and the coverage area of the rainfall are decreased with the reduction of discharge. When we adopted the recommended bucket, the powerhouse and the owner camp were no longer in the rainstorm area under all of the working conditions.

Based on the physical model test, it was found that the distribution of the nappe wind in the axial and vertical directions tended to attenuate, while for recommended bucket, the attenuation degree of the nappe wind was larger than that of other buckets. However, the lateral distribution of the nappe wind varied significantly from one type to another. The nappe wind of the continuous bucket satisfied the normal distribution. The peak value of the nappe wind appeared on the axis of the spillway, and the longer the distance between the measuring point and spillway axis was, the more rapidly the nappe wind decreased. However, for the tongue-shaped bucket, the peak value of the nappe wind appeared on both sides of the spillway axis, which is obviously different from the continuous bucket. For the recommended bucket, as an effect of the bucket type, the peak value of the nappe wind turned to the right side of the spillway axis, and the nappe wind decreased rapidly from right to left.

This paper presents an improved mathematical model and provides a solution for the established water conservancy project which has caused huge economic losses without fully considering the atomization problem. 

## Figures and Tables

**Figure 1 ijerph-16-00316-f001:**
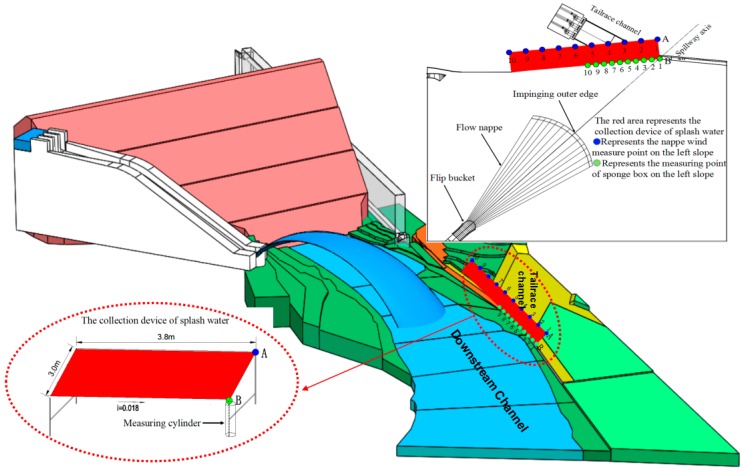
Arrangement of the physical model test.

**Figure 2 ijerph-16-00316-f002:**
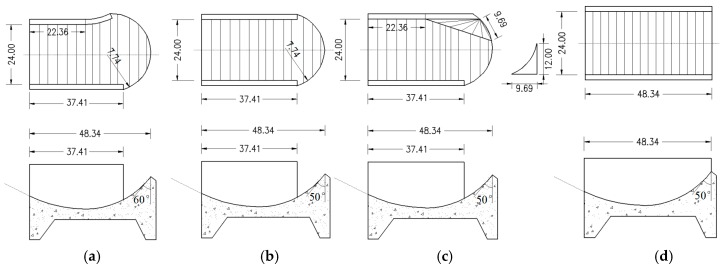
Flip bucket types of the physical model test: (**a**) shape 1; (**b**) shape 2; (**c**) shape 3; (**d**) shape 4; (units: cm).

**Figure 3 ijerph-16-00316-f003:**
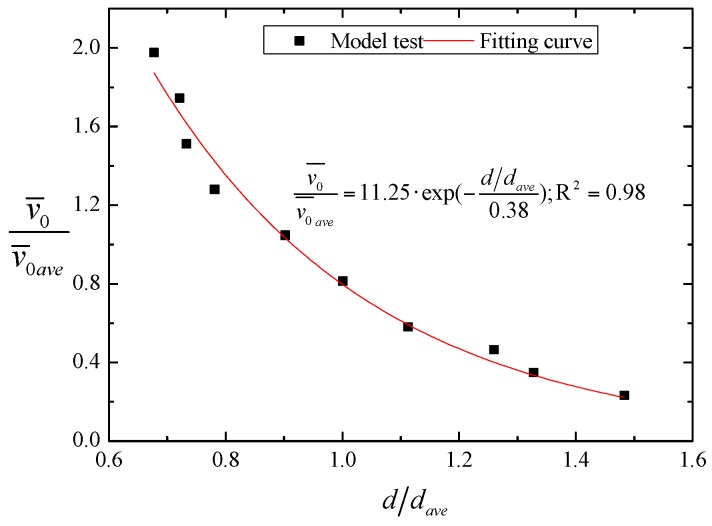
The dimensionless relationship between the droplet diameter and velocity of the physical model test.

**Figure 4 ijerph-16-00316-f004:**
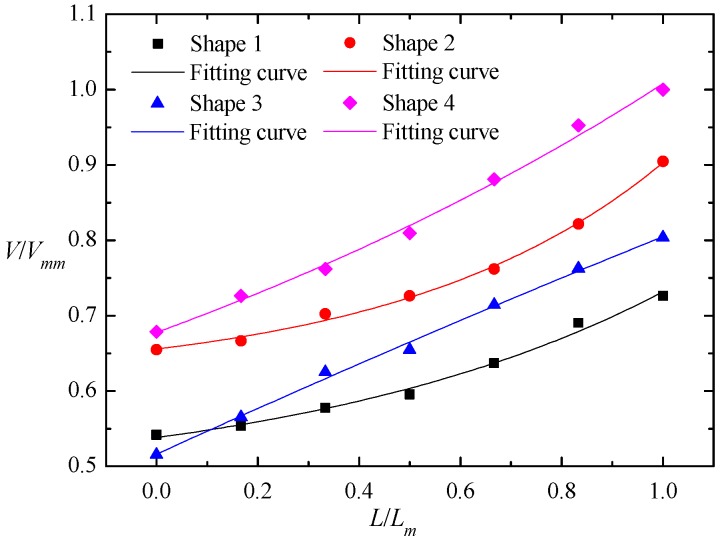
Nappe wind distribution in the longitudinal direction in the physical model test. *V_mm_* is the maximum velocity among four shapes.

**Figure 5 ijerph-16-00316-f005:**
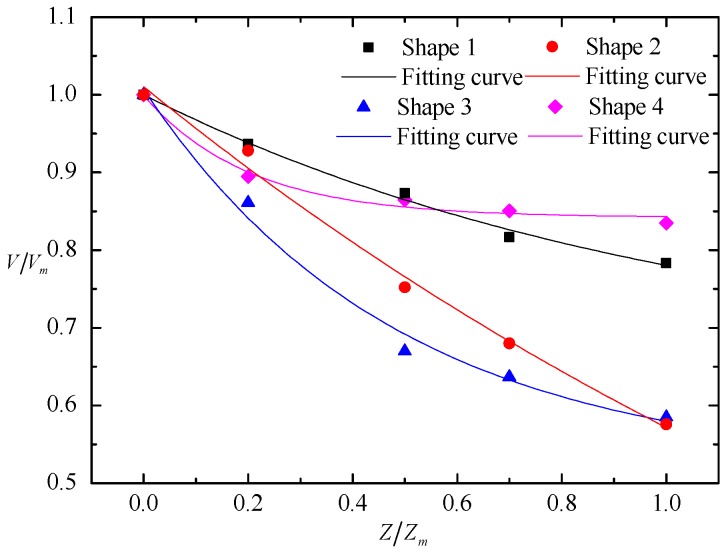
Nappe wind distribution in the vertical direction in the physical model test. *V_m_* is the maximum velocity of each shape.

**Figure 6 ijerph-16-00316-f006:**
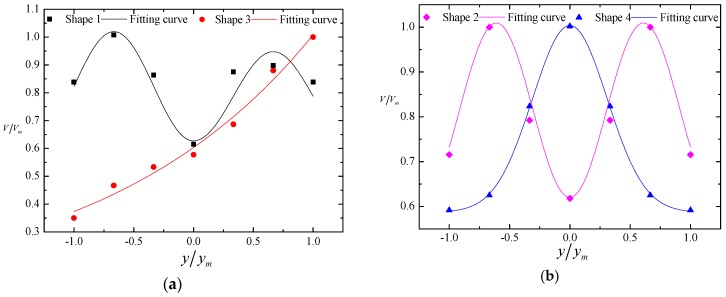
Nappe wind distribution in the lateral direction in the physical model test: (**a**) shape 1 and shape 3; (**b**) shape 2 and shape 4.

**Figure 7 ijerph-16-00316-f007:**
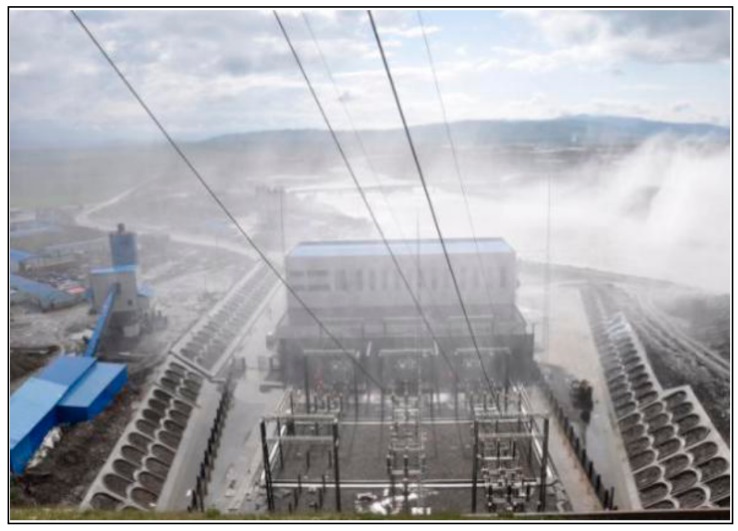
Prototype observation of powerhouse rainfall.

**Figure 8 ijerph-16-00316-f008:**
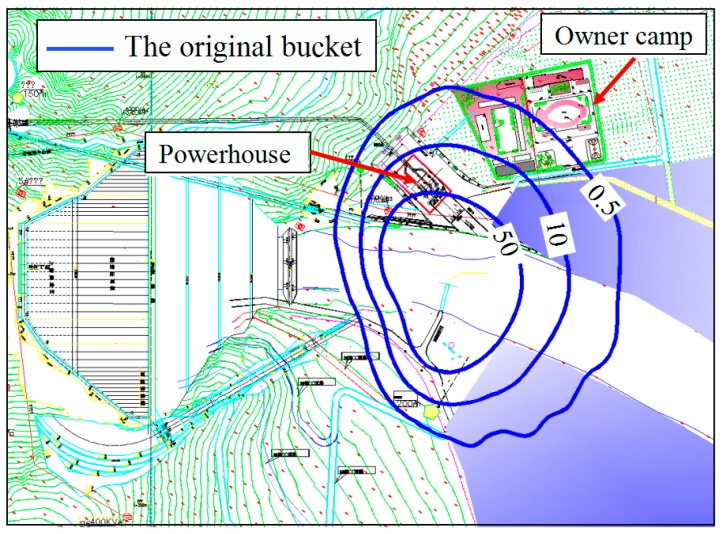
Rainfall intensity distribution of prototypical numerical results (unit: mm *h*^−1^).

**Figure 9 ijerph-16-00316-f009:**
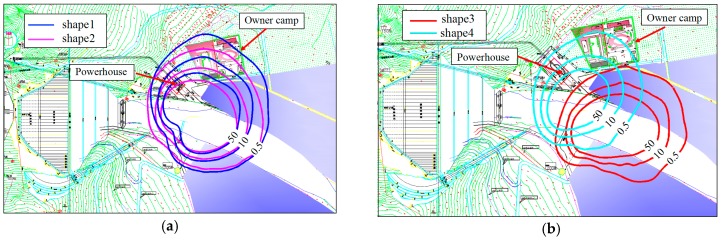
Rainfall intensity distribution of different bucket types of the prototype: (**a**) rainfall intensity distribution of shape 1 and shape 2; (**b**) rainfall intensity distribution of shape 3 and shape 4 (unit: mm *h*^−1^).

**Figure 10 ijerph-16-00316-f010:**
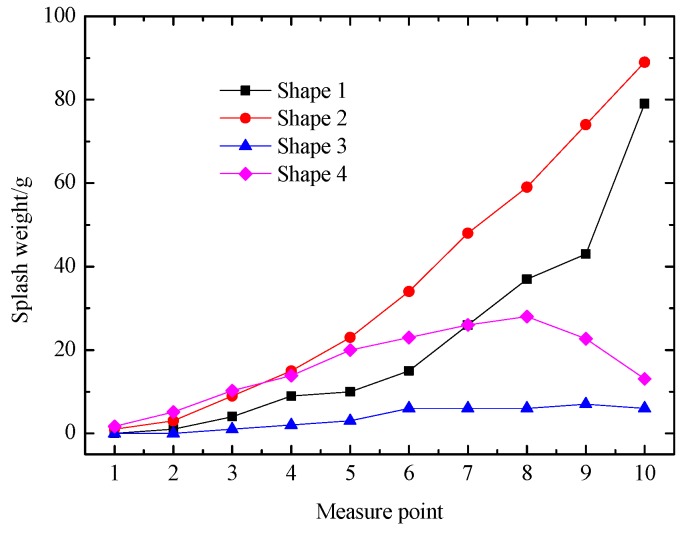
Splash weight at the bottom of the left slope in the physical model test.

**Figure 11 ijerph-16-00316-f011:**
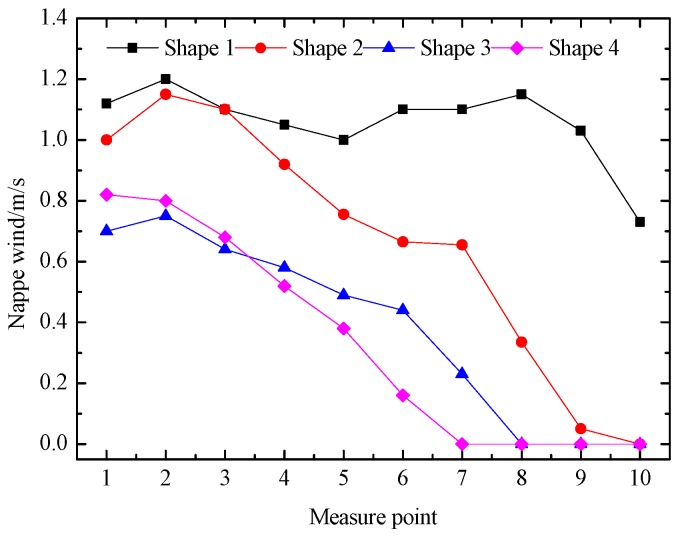
Nappe wind at the upstream slope in the physical model test.

**Figure 12 ijerph-16-00316-f012:**
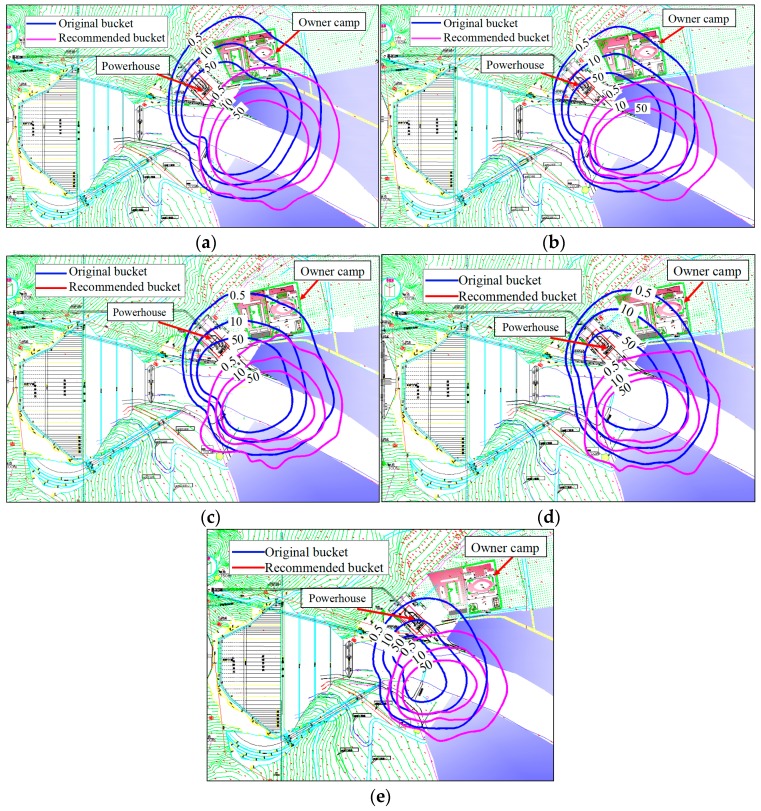
Rainfall intensity distribution of original and recommended buckets with different prototype conditions: (**a**) condition 1; (**b**) condition 2; (**c**) condition 4; (**d**) condition 5; (**e**) condition 7 (units: mm *h*^−1^).

**Figure 13 ijerph-16-00316-f013:**
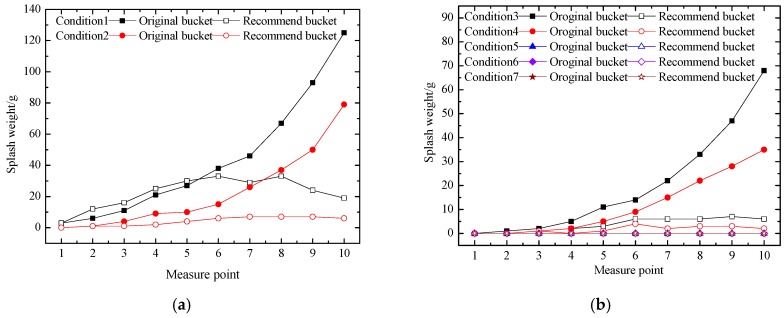
Splash weight at the bottom of the left slope of the physical model test: (**a**) represents the splash weight of condition 1 to condition 2; (**b**) represents the splash weight of condition 3 to condition 7.

**Figure 14 ijerph-16-00316-f014:**
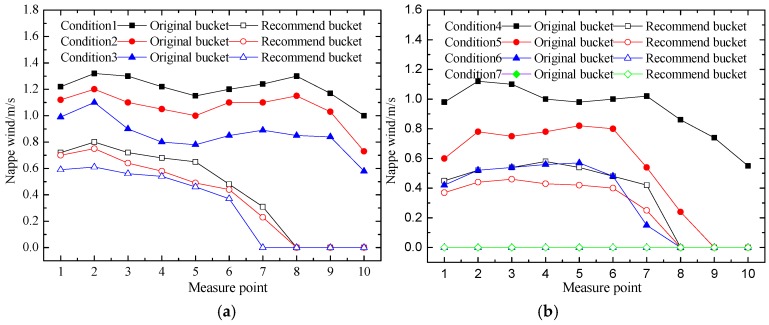
Nappe wind at the upstream slope of the powerhouse of the physical model test; (**a**) represents the nappe wind of condition 1 to condition 3; (**b**) represents the nappe wind of condition 4 to condition 7.

**Table 1 ijerph-16-00316-t001:** Prototypical discharge conditions.

Operating Condition	Height Above Sea Level (M)		Gate Opening Ratio (M)	Discharge (M^3^/S)
	Upstream	Downstream		
1 (Check water level)	3202.38	3090.00	full open	1094.90
2 (Design water level)	3201.70	3090.00	full open	1006.00
3 (Normal water level)	3201.50	3090.00	full open	972.00
4	3201.50	3090.00	50.00%	848.53
5	3201.50	3090.00	25.00%	531.70
6	3201.50	3090.00	10.00%	350.62
7	3201.50	3090.00	5.00%	150.71

Note: The gate opening ratio is defined as the ratio of the gate opening height to the maximum opening height.

**Table 2 ijerph-16-00316-t002:** The weight of splash water for different bucket types in the physical model test.

Bucket Types	Shape 1 (Original Bucket)	Shape 2	Shape 3	Shape 4
Splash weight (g)	4027.0	2802.0	28.0	292.0
Percentage	100%	69.58%	0.69%	7.25%

**Table 3 ijerph-16-00316-t003:** The splash water weight under different conditions in the physical model test.

Operating Condition	1	2	3	4	5	6	7
Original bucket (g)	6335.0	4207.0	3331.0	1246.0	0	0	0
Recommend bucket (g)	240.0	28.0	6.0	0	0	0	0
Percentage	3.79%	0.66%	0.18%	0	0	0	0

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
