# Peer review of "An Improved Empirical Model for Flood Discharge Atomization and Its Application to Optimize the Flip Bucket of the Nazixia Project"

_ijerph, 2019, doi:10.3390/ijerph16030316_

Round 1

Author Response

The Explanation

Manuscript ID: ijerph-410901. Title: “An improved model for flood discharge atomization and its application to optimize the flip bucket of Nazixia project”.

To reviewer: 1 

Dear Editors and Reviewers:

Thanks for your kind letter and for the reviewers’ comments concerning our manuscript. Those comments are helpful for us to revise and improve our paper. We have studied comments carefully and tried our best to revise and improve the manuscript and made great changes in the manuscript according to the reviewers’ good comments. The responds to the reviewer’s comments are as following

Comments to the Author 

 (i):More details should be provided on the physical model and instruments used in this experimental study. More important, the Authors should discuss the negligibility of scale effects in terms of Reynolds and Weber numbers and in simulating water atomization;

Response to reviewer: We are really appreciated for your suggestion. As you said, the description of the physical model and the instruments parameters are incomplete, so we have provided more details about them in the revised version, it is shown as following:

The supplement tank with the length of 10 m, the width of 12 m and the height of 6 m, the downstream channel with the length of 9.5 m, the width of 4.2 m and the height of 0.2 m. The height between the flip bucket and downstream water surface is 0.33 m. The droplets diameter and splashing velocity are measured by the laser raindrop spectrograph (Diameter measurement range:0.1258.5 mm; Velocity measurement range:0.220 m/s; Intensity measurement range: 0.005250 mm/h), which can collect and classify a large number of droplets characteristic parameters based on the droplets diameter.

As you said that the discussion of the scale effects in terms of Reynolds and Weber numbers in simulating flood discharge atomization is necessary, and many researchers have studied that, Wu Shiqiang analyzed the effect of Re and We numbers of flow on the atomization intensity based on a series of physical model tests, it is found that when the scale is small, the values of We and Re are relatively small, the changes in We and Re number have a large influence on rain intensity. With the enlarging of the model scale, the corresponding model flow velocity and discharge increase, as do We and Re, and the influences of surface tension and viscous forces also decrease relatively, when the Weber number is greater than 500, the influences of surface tension and viscous force are relatively weak and have little effect on the test results. Chen Duan divide the splashing droplets into two types: one type is small in number but large in volume, the other is small in volume but large in number; and proposed that different droplets type satisfy different similarity law, which determine the similarity law of rainfall intensity. Although we have made some achievements in the study of the similarity law of flood discharge atomization. the atomization mechanism is extremely complex, and it is vulnerable to the local environmental factors, so further research is necessary on the similarity law of flood discharge atomization. Therefore, in this research, we didn’t use the similarity law to convert the rainfall intensity of the physical model to that of the prototype, but we adopted the qualitative analysis to optimize the shape of flip bucket with the physical model test. As the rainfall intensity distribution around the powerhouse is mainly subject to the splashing water weight and the nappe wind, then we used them as the control indicators to study the rainfall intensity distribution of each flip bucket. The experimental results show that the splashing water weight and the nappe wind of shape 3 around the powerhouse is obviously weaker than that of other buckets, at the same time, the improved theoretical model is conducted to calculate the rainfall intensity distribution of each flip bucket, the powerhouse and the owner camp is no longer in the heavy rain area under all of the working conditions, and the calculation results agree well with the experimental results.

(ii) The Authors present several equations without any discussion/explanation on how they were derived;

Response to reviewer: Thank you for your kindly suggestion,as you said, we have applied a lot of equations and assumptions in this paper, such as the splash flow q,the number n,the random distribution function of the ejection parameters(the diameter, initial velocity, initial elevation angle, initial azimuth angle). As the mechanism of flood discharge atomization is extremely complex,

and it’s a tremendous difficult to use theoretical derivation method to obtain the mechanism. So we conducted a lot of splashing tests and prototype data to obtain the semi-theoretical and semi-empirical formula for calculating the flood discharge atomization, and it was verified to be applicable to predict the atomization of hydropower station by a lot of researchers, therefore, we still adopt these equations in this research. [Lian, J.J.; Li, C.Y.; Liu, F. et al. A prediction method of flood discharge atomization for high dams[J]. Journal of Hydraulic Research, 2014, 52(2), 274-282.;  Liu, F.; Huang, C.Y.; Yang, H. Comparative study of numerical result and field investigation for atomization of high dam. Journal of Hydroelectric Engineering, 2010, 29(1),19-23. (In Chinese); Liu, H.T.; Liu, Z.P.; Xia, Q.F.; et al. Computational model of flood discharge splash in large hydropower stations. Journal of Hydraulic Research, 2015, 53(5), 576-587.]

(iii) More critical insights should be provided on the experimental findings. Several equations from regression analyses are given without any critical comment;

Response to reviewer: Thank you very much for your well-intentioned suggestion. Actually, we have measured parameters of the temperature, humidity, PM2.0 and PM10.0 around the powerhouse, but these parameters are easily influenced by indoor environment, and the sensitivity of the above parameters to the shape of the flip bucket is poor. Therefore, we choose the splashing water weight and the nappe wind as the main control indicators to study the rainfall intensity distribution of each flip bucket.

Due to the mechanism of nappe wind is extremely complex, there are few researches on the distribution of downstream nappe wind, and most of them use generalization function to solve it. However, the spatial distribution of nappe wind is vary from one bucket to another greatly, and the accuracy of the nappe wind has a great influence on the rainfall intensity distribution. Therefore, we have a detailed research on the spatial distribution of the nappe wind.

As the terrain of Nazixia project is flat and the flood discharging only with single hole, so the application of the downstream nappe wind distribution obtained by Nazixia project need further study in the valley area, discharge from surface-bottom outlet condition and the situation of two jets impact in air.

 (iv) Suitable schemes are needed showing the flow nappe and atomization and the related peculiar characteristics. The scheme in Figure 1 is ineffective;

Response to reviewer: Thank you for your kindly suggestion, we have revised it as the following: the flow nappe, atomization and the related peculiar characteristics have been added in Figure 1.

Figure 1. Arrangement of physical model.

(v) the MS requires an extensive revision of the English language and style.

Response to reviewer: Thank you for your kindly suggestion. We have asked foreign experts to revise the grammar problem.

SPECIFIC COMMENTS

[Major Revisions]

[R] Physical model. The Authors used a physical model with a geometric scale 1:50: (i) could the Authors discuss the negligibility of the scale effects mainly in terms of Reynolds and Weber numbers? (ii) Moreover, could the Authors discuss the negligibility of scale effects in simulating water atomization? (iii) (as minor point) At lines 117 and 118, I would write “designed according to Froude similitude.” instead of “designed by the gravity similarity criterion.”; (iv) (still as minor point) At line 128, could the Authors specify the technical meaning of “nappe wind”? (v) At line 140, it reads “Riverbed scouring and silting were measured…”, but could the Authors provide the main characteristics of the mobile bed? More in general, could the Authors discuss the similitude laws for sediment transport? (vi) With reference to Figure 1, what does the photograph on the left represent?

Response to reviewer: We are really appreciated for your suggestion. And we have modified and supplemented them in the revised version.

(i) Wu Shiqiang analyzed the effect of Re and We numbers of flow on the atomization intensity based on a series of physical model tests, it is found that when the scale is small, the values of We and Re are relatively small, the changes in We and Re number have a large influence on rain intensity. With the enlarging of the model scale, the corresponding model flow velocity and discharge increase, as do We and Re, and the influences of surface tension and viscous forces also decrease relatively,

(ii)  The physical model of flood discharge atomization is designed according to gravity similarity criterion, which ensuring the similarity of main physical quantities of water flow first and neglecting the action of the surface tension and viscous force. However, the flood discharge atomization has scale effect, and the similitude law does not follow the gravity similarity criterion. Wu Shiqiang proposed that when the Weber number is greater than 500, the influences of surface tension and viscous force are relatively weak and have little effect on the test results. [Wu, S.Q.; Wu, X.F.; Zhou, H. et al. Analysis and application of the scale effect of flood discharge atomization model[J]. Science China Technological Sciences, 2011, 54(1), 64-71.]

(iii) We have used the “designed according to Froude similitude.” instead of “designed by the gravity similarity criterion in the revised manuscript.

(iv) We have specified the technical meaning of “nappe wind” in the revised manuscript. as it show as following: Through the dragging force between the trajectory nappe and the surrounding air, the surrounding air generate nappe wind by obtaining additional momentum from the high-speed flow nappe.

(v) As for the scouring prediction in model tests, there are three conditions should be satisfied: (1) normal model; (2) hydraulic characteristics of the physical model should satisfy gravity similarity; (3) model sand should satisfy the similarity of sediment mobilization. The similarity method of scouring velocity is often used to simulate the scour of rock foundation, according to the local rock geology, the scouring velocity of rock foundation is determined. Then we use the Isbash formula( represents the scouring velocity(m/s);D represents the sediment grain size(m),K represents the lithology coefficient and its range is from 5 to 7) to calculate the diameter range of the prototype, and then we obtain the diameter range of the physical model.

(vi) The photograph on the left of the previous figure 1 represents the flow regime of the trajectory nappe for the original bucket under designed condition. In the revised manuscript, we have adopted your kindly suggestion about the figure 1, and revised it as the following: the flow nappe, atomization and the related peculiar characteristics have been added in the figure 1.

Figure 1. Arrangement of physical model.

[R] Flip bucket types. With reference to Figure 2, the scheme in Figure2a represents the original bucket of the hydropower station while the remainders represent alternative buckets. Could the Authors justiffy/discuss the ratio behind the shapes of the alternative buckets?

Response to reviewer: Thank you for your kindly suggestion. Usually, there are three types of flip buckets for trajectory energy dissipation :(1) the constant width-type;(2) the expansion-type;(3) contraction-type. As this research, we mainly focus on the established water conservancy project, and we optimize the shape of the bucket with the principle of minimum construction and best effect, therefore, we select the typical flip bucket of each types to study their atomization characteristics. At the same time, we found that the downstream scouring and silting of different buckets increased with the decrease of the bucket angle, and when the bucket angle is too small which can seriously reduce the output of the power station. The splashing droplets and nappe wind near the powerhouse is obviously increased with the increase of the bucket angle, when the bucket angle is too big which can seriously aggravate rainfall intensity. Therefore, we choose the bucket angle as 50o to study the rainfall intensity distribution of different flip bucket.

 [R]The number of splashed water droplets. Could the Authors explain/discuss the equations they provide for the total splash flow q and the number n of splashed water droplets? Overall, this section requires more explanation behind equations and coefficients provided in.

Response to reviewer: Thank you for your kindly suggestion, In this paper, the total splash flow q is expressed as: , and the total splash flow and the numberof splashed water droplets in unit time is calculated as ;C and CM are the water concentration coefficient at any point, and the maximum water concentration coefficient in the same cross-sections, respectively . The relationship between C and CM can be expressed as [Wu, C.G., Yang, Y.S. A study on water concentration distribution in cross-sections along the free jet. J. Hydraulic Eng. 1994, 7, 1–11]:

Figure a Motion of the jet in air

where ; CM andis the maximum and average water concentration coefficient in the same cross-section, respectively. andare the distance from the measurement point to the jet centre in two curvilinear direction. is the thickness of aerated jet flow, and b is the width of the jet flow. is the Froude number, it can be expressed as, represent the thickness of jet flow without aeration.can be expressed as , is the initial Froude number on the outlet of the bucket.

Taking the continuous bucket for example, a rectangle with a length and width  is selected around any point e on the outer boundary of the jet, when the jet impacts at the downstream water surface. The water volume q through the rectangle in unit time can be calculated as.

Figure b The outer boundary as the jet impacting at the downstream water surface

Where , based on the feedback analysis of the prototype and experiment data, , and the coefficient is varies from different buckets, for the continuous bucket ,the value of  ranges from 0.005~0.01, for the silt bucket and twisted bucket, the value of  range from 0.01~0.03. is the thickness of jet flow, which is obtained by physical model test,  is the projection of water droplet velocity in the Z-axis. It is provided that q is completely converted to the splashing water droplets, and the number N of splashing water drops in unit time can be expressed as: ,is the average diameter of water droplets, and the study showed that the value of  is usually selected as a constant that equals to 3mm. it  has been verified in our previous research, therefore we still adopt it in this research.[Lian, J.J.; Li, C.Y.; Liu, F. et al. A prediction method of flood discharge atomization for high dams. Journal of Hydraulic Research, 2014, 52(2), 274-282.]

[R]Stochastic model of splash water droplets. (i) Could the Authors explain/discuss physical reasons according to the diameter d of water droplet and the initial velocity v0 of water droplet would obey to a Gammma distribution (instead, for instance, to a normal distribution)? (ii) At line 225, it reads “we use a laser raindrop spectrograph…”, but the Authors should provide the technical characteristics of this instrument preferably in the section “Physical model”.

(i) Response to reviewer: Thank you for your kindly suggestion, in this paper, the random distribution of the diameter and initial velocity obey to the Gamma distribution. The main reason is that the mechanism of flood discharge atomization is extremely complex, and it’s a tremendous difficult to use theoretical derivation method to obtain the mechanism. So we conducted many splashing tests and prototype data to obtain the semi-theoretical and semi-empirical formula for calculating the flood discharge atomization, and it was verified to be applicable to predict the atomization of a hydropower station by a lot of researchers, in addition, Liu compared the calculated results of the random splash model with the experimental data indicated that the gamma distribution assumption of the diameter and initial velocity can simulate the atomization phenomena very well, therefore we still adopt it in this research [Lian, J.J.; Li, C.Y.; Liu, F. et al. A prediction method of flood discharge atomization for high dams[J]. Journal of Hydraulic Research, 2014, 52(2), 274-282. Liu, H.T.; Liu, Z.P.; Xia, Q.F.; et al. Computational model of flood discharge splash in large hydropower stations. Journal of Hydraulic Research, 2015, 53(5), 576-587. Liu, F.; Huang, C.Y.; Yang, H. Comparative study of numerical result and field investigation for atomization of high dam. Journal of Hydroelectric Engineering, 2010, 29(1),19-23. (In Chinese); Liu, H.T.; Sun, S.K.; Wang, X.S. et al. Study on the distribution of splash intensity during nappe impingement. Journal of Hydrodynamics, 2009, 24(2), 217-223.(In Chinese)].

(ii) Response to reviewer: Thank you for your kindly suggestion, in this paper, the technical characteristics of the laser raindrop spectrograph have been provided as the following: The droplets diameter and splashing velocity are measured by the laser raindrop spectrograph (Diameter measuring range:0.125-8.5 mm; Velocity measuring range:0.2- 20 m/s; Intensity measuring range: 0.005 -250 mm/h)which can collect and classify a large number of droplets characteristic parameters based on the droplets diameter.

[R]Nappe wind and flood discharge. (i) In this section the Authors present several equations from

regression analyses. The coefficients of determination R2are very high (around 0.98), but because the experimental data are very few (typically 5, 6 points). These equations would appear specific and I’m not sure it’s worth providing them! (ii) Could the Authors physically explain/discuss the sinusoidal trends in Figure 6?

(i) Response to reviewer: Thank you for your kindly suggestion, as the diameter of the ultrasonic wind sensor is 30 cm, so we arranged 7 measuring points in the lateral direction with the interval of 50 cm. In addition, we have made repeated measurements to reduce the test error for each measuring point. We found that for the continuous bucket, the nappe wind obeys the normal distribution and the peak value is on the bucket axis, which is consistent with the conclusion of the research. Affected by the types of tongue-shaped bucket, the peak value of the nappe wind is on the both sides of the bucket axis like hump, which is obviously different from that of continuous bucket. The main reason is that for tongue-shaped bucket, the impingement angle of trajectory nappe on the bucket axis direction is larger than that of the both sides, which lead to the horizontal component of nappe wind velocity on the both sides is larger than that of the nappe wind velocity on the bucket axis. While affected by the shape of continuous bucket, the impingement angle of trajectory nappe is unchange, since the width of the impinge edge is small that the plane distribution of nappe wind satisfied normal distribution. [Liu, H.T.; Sun, S.K.; Wang, X.S. et al. Study on the distribution of splash intensity during nappe impingement. Journal of Hydrodynamics, 2009, 24(2), 217-223.(In Chinese)],

(ii) Response to reviewer: Thank you for your kindly suggestion, In this paper, the sinusoidal trends was used to express the lateral distribution of the downstream nappe wind, and it just indicate that the nappe wind value obey to the sinusoidal trends but without any actual physical meanings. it is mainly used to introduced into our mathematical model to better predict the rainfall intensity distribution.

[R] The motion equations of water droplet. Could the Authors explain/discuss how equations from (24) to (27) were derived?

Response to reviewer: Thank you for your kindly suggestion. Firstly, we are sorry for that we missed some references in typesetting, we have added it in the revised manuscript. Actually, in this paper, the equations of motion for the splashing droplets were cited from the research of Lian and Liu. and it was verified to be applicable to predict the atomization of a hydropower station by a lot of researchers, therefore we still adopt it in this research. [Lian, J.J.; Li, C.Y.; Liu, F. et al. A prediction method of flood discharge atomization for high dams[J]. Journal of Hydraulic Research, 2014, 52(2), 274-282. Liu, H.T.; Liu, Z.P.; Xia, Q.F.; et al. Computational model of flood discharge splash in large hydropower stations. Journal of Hydraulic Research, 2015, 53(5), 576-587. Liu, F.; Huang, C.Y.; Yang, H. Comparative study of numerical result and field investigation for atomization of high dam. Journal of Hydroelectric Engineering, 2010, 29(1),19-23. (In Chinese)].

[R] Verification of the improved stochastic model. (i) I believe this title is misleading. It would suggest an interesting comparison of the experimental results with field observations! Instead, if I have understood correctly, this section implies a mere application of the empirical equations presented in Section 2; (ii)Could the Authors explain why they claim their equations constitute an “improved” stochastic model? (iii)More in general could the Author discuss their model in comparison to literature models (if any)?

(i) Response to reviewer: Thank you for your kindly suggestion, in section 3,we combined the improved mathematical model with the physical model to optimize the flip bucket of Nazixia project, the calculation is consistent with experimental results, and we found that shape 3 is the recommend bucket under different discharge condition, actually, this part mainly use two complementary methods (the method of the physical model test and the mathematical model) to optimize the flip bucket of Nazixia project, at the same time, the test results are used to verify the mathematical model.

(ii) Response to reviewer: Thank you for your kindly suggestion. The previous mathematical model in have significant disadvantages as following: (1)( empirical formula has poor accuracy in calculating the hydraulic characteristics (the pitch, width, height and velocity of flow) of the tongue-shaped bucket, curved surface bucket and other extraordinary shaped bucket, which would cause a large error in calculation of the impinging width of water jet; (2) the previous researches didn’t give a detail description on the spatial distribution of nappe wind, but adopting the generalization function to deal with the downstream nappe wind that is unreasonable; (3) the correlation between the diameters and the corresponding velocities of the splashing droplet is neglected. While these factors mentioned above have a great influence on the rainfall intensity and distribution. In order to improve the prediction accuracy of stochastic splash model, a physical model test is conducted to obtain the hydraulic characteristics parameters of impinging outer edge of water jet which is used to form a linear ejection source and it is divided into multiple segments. Furthermore, the spatial distribution of nappe wind is also obtained, and a laser raindrop spectrograph is also installed to obtain the dimensionless relationship between the droplet diameter and splashing velocity. Accordingly, these factors mentioned above are used to improve the mathematical model.

(iii) Response to reviewer: Thank you for your kindly suggestion, In addition to the method of random splashing mathematical model, many scholars also use the BP neural network and a mixture neural network model based on RBF to quantitatively predict the rainfall intensity distribution to simulate the rainfall intensity distribution of flood discharge atomization,however, as the method of neural network needs a lot of training samples, but each practical project has its own particularity, which results in poor generality. In addition, the method don’t have the mechanical background. So we adopt the stochastic mathematical model to predict the rainfall intensity distribution of flood discharge in this research.

[Minor Revisions (but the list could be even longer!)]

[r] Abstract. The last sentence appears incomplete.

Response to reviewer: Thanks for pointing out the mistake in the manuscript, in the revised manuscript, The last sentence of the abstract “And it shows that the atomization influence during the flood discharge can be reduced by using the recommended” have been replaced by the “And it shows that the atomization influence during the flood discharge can be reduced by using the recommended bucket”

[r]  Line 181. It reads “range from”, but it should read “ranges from”.

Response to reviewer: Thanks for pointing out the mistakes in the manuscript, in the revised manuscript,  the “range from” has been replaced by the “ranges from” .

[r]  Line 204. It reads “appear[24]: At”, but it should read “appear [24]: at”.

Response to reviewer: Thanks for pointing out the mistakes in the manuscript, in the revised manuscript, the “appear[24]: At” has been replaced by the “appear [24]: at”.

[r]  Line 206. It reads “plate; Since”, but it should read “plate; since”.

Response to reviewer: Thanks for pointing out the mistakes in the manuscript, in the revised manuscript, the “plate; Since” is replaced by the “plate; since”. 

[r]  Lines 274 and 313. It reads “bucker”, but it should read “bucket”.

Response to reviewer: Thanks for pointing out the mistakes in the manuscript, in the revised manuscript, the “bucker” has been replaced the “bucket”

[r]  Figure 8 caption. This figure caption is incomplete.

Response to reviewer: Thanks for pointing out the mistakes in the manuscript, in the revised manuscript, we have revised it as following:

[r]  Ref. #18. It reads , but it should read “Liu, S.-H.;  Yin, S.-R.; Luo, Q.-S.; Zhou, L.-C. Numerical”, according to the IJERPH Journal manuscript guidelines.

Response to reviewer: Thanks for pointing out the mistakes in the manuscript, in the revised manuscript, the “Liu, S. H., Sun, X. F., & Luo, J.” has been replaced by the Liu, S.H.; Yin, S.R.; Luo, Q.S.; Zhou, L.C. ”

[r]  Ref.  27. It reads “Duan Hong-dong, Liu Shi-he. Rain”, but it should read “Duan, H.-D.; Liu, S.-H.; Luo, Q.-S. Rain; Huang, W. Rain”, according to the IJERPH Journal manuscript guidelines!

Response to reviewer: Thanks for pointing out the mistakes in the manuscript, in the revised manuscript, the “Duan Hong-dong, Liu Shi-he. Rain” has been replaced by the “Duan, H.D.; Liu, S.H.; Luo, Q.S. Rain; Huang, W. Rain”;

[r]  References. More in general, the Authors should carefully check all the references controlling that their style agrees with the IJERPH Journal manuscript guidelines!

Response to reviewer: Thanks for pointing out the mistakes in the manuscript, in the revised manuscript, and we have carefully checked all the references controlling that their style agrees with the IJERPH Journal manuscript guidelines.

Reviewer 2 Report

1.     The manuscript presents an improved model for flood discharge atomization and its application to optimize the flip bucket of Nazixia project, China, which is interesting. The subject addressed is within the scope of the journal.

2.     However, the manuscript, in its present form, contains several weaknesses. Appropriate revisions to the following points should be undertaken in order to justify recommendation for publication.

3.     For readers to quickly catch your contribution, it would be better to highlight major difficulties and challenges, and your original achievements to overcome them, in a clearer way in abstract and introduction.

4.     It is shown in the reference list that the authors have several publications in this field. This raises some concerns regarding the potential overlap with their previous works. The authors should explicitly state the novel contribution of this work, the similarities and the differences of this work with their previous publications.

5.     It is mentioned in p.3 that Nazixia Hydropower Station is adopted as the case study. What are other feasible alternatives? What are the advantages of adopting this particular case study over others in this case? How will this affect the results? The authors should provide more details on this.

6.     It is mentioned in p.4 that 4 bucket types are adopted in this study. What are the other feasible alternatives? What are the advantages of adopting these particular types over others in this case? How will this affect the results? More details should be furnished.

7.     It is mentioned in p.6 that a specific equation is adopted for the number of splashed water droplets in unit time. What are other feasible alternatives? What are the advantages of adopting this particular equation over others in this case? How will this affect the results? The authors should provide more details on this.

8.     It is mentioned in p.6 that [21,26] is adopted for the water concentration. What are other feasible alternatives? What are the advantages of adopting this particular approach over others in this case? How will this affect the results? The authors should provide more details on this.

9.     It is mentioned in p.7 that Gamma distribution is adopted for the diameter d of water droplet. What are other feasible alternatives? What are the advantages of adopting this particular distribution over others in this case? How will this affect the results? The authors should provide more details on this.

10.  It is mentioned in p.7 that a laser raindrop spectrograph is adopted to obtain the dimensionless relationship between the droplet diameter and splashing velocity. What are other feasible alternatives? What are the advantages of adopting this particular tool over others in this case? How will this affect the results? The authors should provide more details on this.

11.  It is mentioned in p.8 that the normal distribution is adopted for the initial azimuth angleof water droplet. What are other feasible alternatives? What are the advantages of adopting this particular distribution over others in this case? How will this affect the results? The authors should provide more details on this.

12.  It is mentioned in p.8 that the ultrasonic wind sensor is adopted to measure the spatial distribution of the nappe wind. What are other feasible alternatives? What are the advantages of adopting this particular tool over others in this case? How will this affect the results? The authors should provide more details on this.

13.  It is mentioned in p.8 that “…In the spillway axial direction, the nappe wind of shape 1 is smaller than that of all other shapes. The main reason is…” More justification should be furnished on this issue.

14.  It is mentioned in p.10 that the Monte-Carlo theory is adopted to generate initial elevation angle and azimuth angle. What are other feasible alternatives? What are the advantages of adopting this particular approach over others in this case? How will this affect the results? The authors should provide more details on this.

15.  It is mentioned in p.11 that the Runge-Kutta method is adopted to calculate the motion differential equations. What are other feasible alternatives? What are the advantages of adopting this particular method over others in this case? How will this affect the results? The authors should provide more details on this.

16.  It is mentioned in p.13 that “…the splash weight of shape 2, shape 3 and shape 4 only accounts for 69.58%,0.69% and 7.25% that of the shape 1, respectively, the main reason is tha …” More justification should be furnished on this issue.

17.  It is mentioned in p.14 that “…The nappe wind of shape 3 and shape 4 attenuate rapidly, which is mainly because …” More justification should be furnished on this issue.

18.  It is mentioned in p.14 that an optimized stochastic splash model is adopted to calculate the rainfall intensity distribution. What are other feasible alternatives? What are the advantages of adopting this particular model over others in this case? How will this affect the results? The authors should provide more details on this.

19.  Some key parameters are not mentioned. The rationale on the choice of the particular set of parameters should be explained with more details. Have the authors experimented with other sets of values? What are the sensitivities of these parameters on the results?

20.  Some assumptions are stated in various sections. Justifications should be provided on these assumptions. Evaluation on how they will affect the results should be made.

21.  The discussion section in the present form is relatively weak and should be strengthened with more details and justifications.

22.  There are some occasional grammatical problems within the text. It may need the attention of someone fluent in English language to enhance the readability.

23.  Moreover, the manuscript could be substantially improved by relying and citing more on recent literatures about contemporary real-life case studies of optimization techniques in water resources engineering such as the followings:

l   Fotovatikhah, F., et al., “Survey of Computational Intelligence as Basis to Big Flood Management: Challenges, research directions and Future Work,” Engineering Applications of Computational Fluid Mechanics 12 (1): 411-437 2018.

l   Taormina, R., et al., “Neural network river forecasting through baseflow separation and binary-coded swarm optimization”, Journal of Hydrology 529 (3): 1788-1797 2015.

l   Wu, C.L., et al., “Rainfall-Runoff Modeling Using Artificial Neural Network Coupled with Singular Spectrum Analysis”, Journal of Hydrology 399 (3-4): 394-409 2011.

l   Wang, W.C., et al., “Improved annual rainfall-runoff forecasting using PSO-SVM model based on EEMD,” Journal of Hydroinformatics 15 (4): 1377-1390 2013.

l   Cheng, C.T., et al., “Flood control management system for reservoirs,” Environmental Modeling & Software 19 (12): 1141-1150 2004.

l   Chau, K.W., et al., “Use of Meta-Heuristic Techniques in Rainfall-Runoff Modelling” Water 9(3): article no. 186, 6p 2017.

24.  Some inconsistencies and minor errors that needed attention are:

l   Replace “…When the jet impact with…” with “…When the jet impacts with…” in line 203 of p.6

l   Replace “…of nappe wind is extremely complex…” with “…of nappe wind being extremely complex…” in line 242 of p.8

l   Replace “…fitting, As it shown in…” with “…fitting. As is shown in…” in line 281 of p.9

l   Replace “…Base on the random…” with “…Based on the random…” in line 321 of p.10

l   Replace “…can not be reduced…” with “…cannot be reduced…” in line 356 of p.12

l   and many more…

25.  In the conclusion section, the limitations of this study, suggested improvements of this work and future directions should be highlighted.

Author Response

The Explanation

Manuscript ID: ijerph-410901. Title: “An improved model for flood discharge atomization and its application to optimize the flip bucket of Nazixia project”.

To reviewer: 2 

Dear Editors and Reviewers:

Thanks for your kind letter and for the reviewers’ comments concerning our manuscript. Those comments are helpful for us to revise and improve our paper. We have studied comments carefully and tried our best to revise and improve the manuscript and made great changes in the manuscript according to the reviewers’ good comments. The responds to the reviewer’s comments are as following.

Point 1: For readers to quickly catch your contribution, it would be better to highlight major difficulties and challenges, and your original achievements to overcome them, in a clearer way in abstract and introduction.

Response to reviewer: We are really appreciated for your suggestion, as you suggested that we should highlight the major difficulties and challenges of the previous stochastic mathematical model of flood discharge atomization and emphasize our original achievements to improve the mathematical mode. in the revised manuscript, we have added the description about that as following:

Abstract: The flood discharge atomization is a serious challenge that threatens the daily life of the residents around the dam area as well as the safety of water conservancy project. The previous mathematical models have the following shortcomings: ① the empirical formula has poor accuracy in calculating the hydraulic characteristics of the extraordinary shaped bucket; ② neglecting the correlation between the diameter and the corresponding velocity of the splashing droplet is not conform to the actual condition; ③ adopting the generalization function to deal with the downstream nappe wind is unreasonable. This research aims at addressing the above deficiencies, a physical model test with four type flip buckets is conducted to obtain the hydraulic parameters of the impinging outer edge of water jet, the relationship of the splashing droplet diameter and its corresponding velocity, and the spatial distribution of downstream nappe wind, these factors mentioned above are introduced to improve the accuracy of the mathematical mode.

Introduction However, the previous mathematical model have significant disadvantages as the following: (1) the empirical formula has poor accuracy in calculating the hydraulic characteristics (the pitch, width, height and velocity of flow) of the tongue-shaped bucket, curved surface bucket and other extraordinary shaped bucket, which would cause a large error in calculation of the impinging width of water jet; (2) the previous researches didn't give a detail description on the spatial distribution of nappe wind but using the generalization function; (3) the correlation between the diameters and the corresponding velocities of the splashing droplet is neglected. while these factors mentioned above have a great influence on the rainfall intensity and distribution. To improve the prediction accuracy of mathematical splash model, a physical model test is conducted to obtain the hydraulic characteristics parameters of impinging outer edge of water jet. which is used to form a linear ejection source, and it is divided into multiple segments. Furthermore, we use a laser raindrop spectrograph to obtain the dimensionless relationship between the droplet diameter and splashing velocity, and the ultrasonic wind sensor is conducted to measure the spatial distribution of the nappe wind. Accordingly, the mathematical model for atomization is improved by the above factors.

 Point 2: It is shown in the reference list that the authors have several publications in this field. This raises some concerns regarding the potential overlap with their previous works. The authors should explicitly state the novel contribution of this work, the similarities and the differences of this work with their previous publications.

Response to reviewer: Thank you for your kindly suggestion. This research aim at solving the shortcomings of the previous theoretical model: Empirical formula has poor accuracy in calculating the hydraulic characteristics (the pitch, width, height and velocity of flow) of the tongue-shaped bucket, curved surface bucket and other extraordinary shaped bucket, which would cause a large error in calculation of the impinging width of water jet; It cannot give a detail description on the spatial distribution of nappe wind; The correlation between the diameters and the corresponding velocities of the splashing droplet is neglected. While these factors mentioned above have a great influence on the rainfall intensity and distribution.Therefore, based on the previous research, the optimization is carried out from the following aspects:

(1) In the previous research, the empirical formula was always used to calculate the hydraulic characteristics of the pitch and the width of impinging outer edge of water jet, but for the special shaped bucket, such as the tongue-shaped bucket and curved surface bucket, the flow regime is extremely complicated, and the empirical formula is difficult to calculate the width and impact location of water jet impingement accurately. while the pitch and the impinging width of water jet have a great influence on the rainfall intensity distribution. Therefore, we used the physical model test to obtain the hydraulic parameters of impinging outer edge of water jet which forms a linear ejection source and divide it into multiple segments.

(2) In the previous stochastic mathematical model research, velocity and diameter of water droplets were taken as independent variable respectively which is unreasonable, because there is a certain correlation between splashing droplet diameter and its velocity obtained from the physical model analysis. With the droplet diameter increasing, the ejection required larger initial impulse. So for the large diameter droplets, the splash velocity and splash area is relatively small, and conversely the small diameter droplets have high velocity and wide splash area. and the relationship of velocity and diameter have a great influence on simulating of the rainfall intensity distribution. Therefore,in this research,we use a laser raindrop spectrograph to obtain the dimensionless relationship between the droplet diameter and splashing velocity.

(3) Due to the mechanism of nappe wind is extremely complex, there are few research on the distribution of nappe wind, and most previous stochastic mathematical model of flood discharge use generalization function to solve it. However, the spatial distribution of nappe wind is vary from one bucket to another greatly, so using the generalization function is unreasonable. And the accuracy of the nappe wind has a great influence on the rainfall intensity distribution. Therefore, we use the ultrasonic wind sensor to measure the spatial distribution of the nappe wind.

 [Lian, J.J.; Li, C.Y.; Liu, F. et al. A prediction method of flood discharge atomization for high dams. Journal of Hydraulic Research, 2014, 52(2), 274-282.;  Liu, F.; Huang, C.Y.; Yang, H. Comparative study of numerical result and field investigation for atomization of high dam. Journal of Hydroelectric Engineering, 2010, 29(1),19-23. (In Chinese).]

 Point 3: It is mentioned in p.3 that Nazixia Hydropower Station is adopted as the case study. What are other feasible alternatives? What are the advantages of adopting this particular case study over others in this case? How will this affect the results? The authors should provide more details on this.

Response to reviewer: Thank you for your kindly suggestion, In this research ,we choose Nazixia as a typical case that is mainly because this project has caused huge economic losses without fully considering the atomization problem. In the process of design and construction, the powerhouse was located at the axis of spillway which caused the powerhouse is in the heavy rain area, and threatens the daily life of the residents around the dam area as well as the safety of water conservancy project. In addition, Nazixia adopted the trajectory energy dissipation which is widely used in water conservancy project, so it has reference value for similar projects.

 Point 4:  It is mentioned in p.4 that 4 bucket types are adopted in this study. What are the other feasible alternatives? What are the advantages of adopting these particular types over others in this case? How will this affect the results? More details should be furnished.

Response to reviewer: Thank you for your kindly suggestion, Usually, there are three types of flip buckets for trajectory energy dissipation :(1) the constant width-type;(2) the expansion-type;(3) contraction-type. As this research, we mainly focus on the established water conservancy project, and we optimize the shape of the bucket with the principle of minimum construction and best effect, therefore, we select the typical flip bucket of each types to study their atomization characteristics. At the same time, we found that the downstream scouring and silting of different buckets increased with the decrease of the bucket angle, and when the bucket angle is too small which can seriously reduce the output of the power station. The splashing droplets and nappe wind near the powerhouse is obviously increased with the increase of the bucket angle, when the bucket angle is too big which can seriously aggravate rainfall intensity. Therefore, we choose the bucket angle as 50o to study the rainfall intensity distribution of different flip bucket. We have added more explanation in the revised version.

 Point 5:  It is mentioned in p.6 that a specific equation is adopted for the number of splashed water droplets in unit time. What are other feasible alternatives? What are the advantages of adopting this particular equation over others in this case? How will this affect the results? The authors should provide more details on this.

Response to reviewer: Thank you for your kindly suggestion, many scholars also use the BP neural network and a mixture neural network model based on RBF to quantitatively predict the rainfall intensity distribution to simulate the rainfall intensity distribution of flood discharge atomization,however,as the method of neural network needs a lot of training samples, but each practical project has its own particularity, which results in poor generality. In addition, the method dont have the mechanical background, therefore, in this paper, we based on the feedback analysis of the prototype and experiment data to obtain this specific equation for the number of splashed water droplets in unit time, the more detail for the formulation derivation is show as following:

in this paper, the total splash flow q is expressed as: , and the total splash flow and the numberof splashed water droplets in unit time is calculated as ; C and CM are the water concentration coefficient at any point, and the maximum water concentration coefficient in the same cross-sections, respectively . The relationship between C and CM can be expressed as [Wu, C.G.; Yang, Y.S. A study on water concentration distribution in cross-sections along the free jet. J. Hydraulic Eng. 1994, 7, 1–11]:

Figure a Splashing number of water drops where (a) motion of the jet in air and

where ; CM andis the maximum and average water concentration coefficient in the same cross-section, respectively. andare the distance from the measurement point to the jet centre in two curvilinear direction. is the thickness of aerated jet flow, and b is the width of the jet flow. is the Froude number, it can be expressed as, represent the thickness of jet flow without aeration.can be expressed as , is the initial Froude number on the outlet of the bucket.

Taking the continuous bucket for example, a rectangle with a length and width  is selected around any point e on the outer boundary of the jet, when the jet impacts at the downstream water surface. The water volume q through the rectangle in unit time can be calculated as.

Figure b The outer boundary as the jet impacting at the downstream water surface

Where , based on the feedback analysis of the prototype and experiment data, , and the coefficient is varies from different buckets, for the continuous bucket ,the value of  ranges from 0.0050.01, for the silt bucket and twisted bucket, the value of  range from 0.010.03. is the thickness of jet flow, which is obtained by physical model test,  is the projection of water droplet velocity in the Z-axis. It is provided that q is completely converted to the splashing water droplets, and the number N of splashing water dropsin unit time can be expressed as: ,is the average diameter of water droplets, and the study showed that the value of  is usually selected as a constant that equals to 3mm. As it has been verified in our previous research, therefore we still adopt it in this research. [Lian, J.J.; Li, C.Y.; Liu, F. et al. A prediction method of flood discharge atomization for high dams. Journal of Hydraulic Research, 2014, 52(2), 274-282. Liu, F.; Huang, C.Y.; Yang, H. Comparative study of numerical result and field investigation for atomization of high dam. Journal of Hydroelectric Engineering, 2010, 29(1),19-23. (In Chinese).]

 Point 6: It is mentioned in p.6 that [21,26] is adopted for the water concentration. What are other feasible alternatives? What are the advantages of adopting this particular approach over others in this case? How will this affect the results? The authors should provide more details on this.

Response to reviewer: Thank you for your kindly suggestion, C and CM are the water concentration coefficient at any point, and the maximum water concentration coefficient in the same cross-sections, respectively . The relationship between C and CM can be expressed as [[Wu, C.G.; Yang, Y.S. A study on water concentration distribution in cross-sections along the free jet. J. Hydraulic Eng. 1994, 7, 1–11]:

Figure a Splashing number of water drops where (a) motion of the jet in air and

where ; CM andis the maximum and average water concentration coefficient in the same cross-section, respectively. andare the distance from the measurement point to the jet centre in two curvilinear direction. is the thickness of aerated jet flow, and b is the width of the jet flow. is the Froude number, it can be expressed as, represent the thickness of jet flow without aeration.can be expressed as , is the initial Froude number on the outlet of the bucket. In our previous study, we found it feasible to adopt this formula ,which was verified by prototype observation and physical model test. therefore we still adopt it in this research. [Lian, J.J.; Li, C.Y.; Liu, F. et al. A prediction method of flood discharge atomization for high dams[J]. Journal of Hydraulic Research, 2014, 52(2), 274-282. Liu, F.; Huang, C.Y.; Yang, H. Comparative study of numerical result and field investigation for atomization of high dam. Journal of Hydroelectric Engineering, 2010, 29(1),19-23. (In Chinese).]

 Point 7:  It is mentioned in p.7 that Gamma distribution is adopted for the diameter d of water droplet. What are other feasible alternatives? What are the advantages of adopting this particular distribution over others in this case? How will this affect the results? The authors should provide more details on this.

Response to reviewer: Thank you for your kindly suggestion, many scholars also use the BP neural network and a mixture neural network model based on RBF to quantitatively predict the rainfall intensity distribution to simulate the rainfall intensity distribution of flood discharge atomization,however,as the method of neural network needs a lot of training samples, but each practical project has its own particularity, which results in poor generality. In addition, the method don’t have the mechanical background. in this paper, we based on the feedback analysis of the prototype and experiment data to obtain Gamma distribution for the diameter of water droplets.

In this paper, the random distribution of the diameter obey to the Gamma distribution. The main reason is that the mechanism of flood discharge atomization is extremely complex, and it’s a tremendous difficult to use theoretical derivation method to obtain the mechanism. So we conducted a lot of splashing tests and prototype data to obtain the semi-theoretical and semi-empirical formula for calculating the flood discharge atomization, and it was verified to be applicable to predict the atomization of a hydropower station by a lot of researchers, in addition, Liu compared the calculated results of the random splash model with the experimental data indicated that the gamma distribution assumption of the diameter and initial velocity can simulate the atomization phenomena very well. Therefore, we still use these equations in this paper. [Lian, J.J.; Li, C.Y.; Liu, F. et al. A prediction method of flood discharge atomization for high dams[J]. Journal of Hydraulic Research, 2014, 52(2), 274-282. Liu, H.T.; Liu, Z.P.; Xia, Q.F.; et al. Computational model of flood discharge splash in large hydropower stations. Journal of Hydraulic Research, 2015, 53(5), 576-587. Liu, F.; Huang, C.Y.; Yang, H. Comparative study of numerical result and field investigation for atomization of high dam. Journal of Hydroelectric Engineering, 2010, 29(1),19-23. (In Chinese). Liu, H.T.; Sun, S.K.; Wang, X.S. et al. Study on the distribution of splash intensity during nappe impingement. Journal of Hydrodynamics, 2009, 24(2), 217-223.(In Chinese)]

 Point 8: It is mentioned in p.7 that a laser raindrop spectrograph is adopted to obtain the dimensionless relationship between the droplet diameter and splashing velocity. What are other feasible alternatives? What are the advantages of adopting this particular tool over others in this case? How will this affect the results? The authors should provide more details on this.

Response to reviewer: Thank you for your kindly suggestion. The stain filter paper and the laser raindrop spectrograph are usually used to measure the parameters of water droplets. But the stain filter paper method is often complicated for statistical classification of droplets diameter which results in time-consuming, poor real-time ability and poor accuracy. The laser raindrop spectrograph as a new method which can overcome the above shortcomings, and it can measure the total numbers, diameter size and velocity of the droplet. It is superior to measure small objects with the minimum diameter of 0.16 mm.

 Point 9: It is mentioned in p.8 that the normal distribution is adopted for the initial azimuth angle of water droplet. What are other feasible alternatives? What are the advantages of adopting this particular distribution over others in this case? How will this affect the results? The authors should provide more details on this.

Response to reviewer: Thank you for your kindly suggestion. In this paper, the random distribution of the initial azimuth angle obey to the normal distribution. As the mechanism of flood discharge atomization is extremely complex, and it’s a tremendous difficult to use theoretical derivation method to obtain the mechanism. So we conducted a lot of splashing test and prototype data to obtain the semi-theoretical and semi-empirical fomula for calculating the flood discharge atomization, and it was verified to be applicable to predict the atomization of a hydropower station by a lot of researchers. Besides, Liu compared the calculated results of the random splash model with the experimental data indicated that the normal distribution assumption of the initial azimuth angle can simulate the atomization phenomena very well. Therefore, we still adopt these equations in this paper. [Lian, J.J.; Li, C.Y.; Liu, F. et al. A prediction method of flood discharge atomization for high dams[J]. Journal of Hydraulic Research, 2014, 52(2), 274-282. Liu, H.T.; Liu, Z.P.; Xia, Q.F.; et al. Computational model of flood discharge splash in large hydropower stations. Journal of Hydraulic Research, 2015, 53(5), 576-587. Liu, F.; Huang, C.Y.; Yang, H. Comparative study of numerical result and field investigation for atomization of high dam. Journal of Hydroelectric Engineering, 2010, 29(1),19-23. (In Chinese). Liu, H.T.; Sun, S.K.; Wang, X.S. et al. Study on the distribution of splash intensity during nappe impingement. Journal of Hydrodynamics, 2009, 24(2), 217-223.(In Chinese)]

Point 10:It is mentioned in p.8 that the ultrasonic wind sensor is adopted to measure the spatial distribution of the nappe wind. What are other feasible alternatives? What are the advantages of adopting this particular tool over others in this case? How will this affect the results? The authors should provide more details on this.

Response to reviewer: Thank you for your kindly suggestion. In this paper, the nappe wind of each measuring point is measured by QYCG-09 ultrasonic wind speed sensor. The ultrasound wind speed and direction sensor uses the influence of air flow (wind) to measure the wind speed. The speed of sound propagation in the air is superimposed with the velocity of air flow in the wind direction. If the direction of ultrasonic propagation is the same as the wind direction, its speed will be accelerated, conversely, contrary to the direction of the wind, it will slow down. Therefore, under the fixed detection conditions, the speed of ultrasonic propagation in the air can correspond to the wind speed function and the accurate wind velocity can be obtained by calculation. therefore, in this paper, we use QYCG-09 ultrasonic wind speed sensor to measure the downstream wind.

 Point 11:  It is mentioned in p.8 that “…In the spillway axial direction, the nappe wind of shape 1 is smaller than that of all other shapes. The main reason is…” More justification should be furnished on this issue.

Response to reviewer: Thank you for your kindly suggestion. as you suggested that the justification of“In the spillway axial direction, the nappe wind of shape 1 is smaller than that of other shapes”should be strengthened. In the revised manuscript we have strengthened them as following: In the spillway axial direction, the nappe wind of shape 1 is smaller than that of other shapes. The main reason is that the nappe wind is attenuated by the action of viscous force and frictional force, as the bucket angle of shape 1 is bigger than others, which caused the jet trajectory length is shorter than others, and the distance between the measuring point and impinging position of jet as well as the effecting time of the viscous force and frictional force is longer than others. Therefore, the nappe wind of shape 1 is smallest among the four types.

 Point 12:It is mentioned in p.10 that the Monte-Carlo theory is adopted to generate initial elevation angle and azimuth angle. What are other feasible alternatives? What are the advantages of adopting this particular approach over others in this case? How will this affect the results? The authors should provide more details on this.

Response to reviewer: Thank you for your kindly suggestion. Firstly, we are sorry for that we missed some references in typesetting, we have added it in the revised manuscript. Monte-Carlo method is based the probability and statistics analysis, using the random sampling method as it main means, and calculating some parameters through statistical experiments. It is verified to be applicable to predict the atomization of a hydropower in previous study. Therefore, we still adopt this method to generate the pseudo random numbers for the diameter, initial velocity, initial elevation angle and azimuth angle in this paper. [Zhang, H.; Lian J.J.; Liu J.K. Monte-Carlo Method for Calculating a Class of Stochastic Differential Equation. Journal of Tianjin University: 2003, 36(4), 430-433. (In Chinese). 13.Lian, J.J.; Li, C.Y.; Liu, F. et al. A prediction method of flood discharge atomization for high dams. Journal of Hydraulic Research, 2014, 52(2), 274-282.]

 Point 13: It is mentioned in p.11 that the Runge-Kutta method is adopted to calculate the motion differential equations. What are other feasible alternatives? What are the advantages of adopting this particular method over others in this case? How will this affect the results? The authors should provide more details on this.

Response to reviewer: Thank you for your kindly suggestion. Firstly, we are sorry for that we missed some references in typesetting, we have added it in the revised manuscript. The Runge-Kutta method is one of the methods which is used to calculate the nonlinear partial differential equations, and it is verified to be applicable to calculate the motion differential equations. Therefore, we still adopt this method in this paper. [Zhang, H.; Lian J.J.; Liu J.K. Monte-Carlo Method for Calculating a Class of Stochastic Differential Equation. Journal of Tianjin University: 2003, 36(4), 430-433. (In Chinese). Lian, J.J.; Li, C.Y.; Liu, F. et al. A prediction method of flood discharge atomization for high dams. Journal of Hydraulic Research, 2014, 52(2), 274-282.]

 Point 14: It is mentioned in p.13 that “…the splash weight of shape 2, shape 3 and shape 4 only accounts for 69.58%,0.69% and 7.25% that of the shape 1, respectively, the main reason is tha …” More justification should be furnished on this issue.

Response to review: Thank you for your kindly suggestion. As you suggested that the reason of“the splash weight of shape 2, shape 3 and shape 4 only accounts for 69.58%,0.69% and 7.25% that of the shape 1”should be strengthened. In the revised manuscript we have strengthened them as following: the main reason is that shape 1 and shape 2 are tongue-shaped bucket, compared with shape 3(skew bucket ) and shape 4 (continuous bucket), the lateral diffusion of the aerated jet is more intensive, and the width of the impinging edge is relatively wide which result in the range of splashing droplets is broader, so the splash weight of shape 1 and shape 2 are obviously larger. For shape 3 and shape 4, as the effect of the bucket type, the lateral diffusion degree of the aerated jet and the width of the impinging edge are reduced, especially for the shape 3, affected by the curved surface that is attached to the left wall, the water jet and the impact point turn to the right side of the spillway axis, which make the shape 3 has the most advantage in reducing the splash weight.

Point 15: It is mentioned in p.14 that “…The nappe wind of shape 3 and shape 4 attenuate rapidly, which is mainly because …” More justification should be furnished on this issue.

Response to review: Thank you for your kindly suggestion. As you suggested that the reason of“The nappe wind of shape 3 and shape 4 attenuate rapidly”should be strengthened. In the revised manuscript we have strengthened them as following: it is mainly because the trajectory nappe of shape 3 and shape 4 is more contracted, and the impinging outer edge of shape 3 and shape 4 is obviously smaller than that of shape 1 and shape 2, which makes the action range of trajectory nappe is relatively small, so the nappe wind of shape 3 and shape 4 attenuate more rapidly than shape 1 and shape 2.

 Point 16: It is mentioned in p.14 that an optimized stochastic splash model is adopted to calculate the rainfall intensity distribution. What are other feasible alternatives? What are the advantages of adopting this particular model over others in this case? How will this affect the results? The authors should provide more details on this.

Response to reviewer: Thank you for your kindly suggestion, many scholars also use the BP neural network and a mixture neural network model based on RBF to quantitatively predict the rainfall intensity distribution to simulate the rainfall intensity distribution of flood discharge atomization, however, as the method of neural network needs a lot of training samples, but each practical project has its own particularity, which results in poor generality. In addition, the method doesn’t have the mechanical background. therefore, In this paper, we adopt the semi-theoretical and semi-empirical formula for calculating the flood discharge atomization, and it was verified to be applicable to predict the atomization of hydropower station by a lot of researchers. [Lian, J.J.; Li, C.Y.; Liu, F. et al. A prediction method of flood discharge atomization for high dams. Journal of Hydraulic Research, 2014, 52(2), 274-282.;  Liu, F.; Huang, C.Y.; Yang, H. Comparative study of numerical result and field investigation for atomization of high dam. Journal of Hydroelectric Engineering, 2010, 29(1),19-23. (In Chinese); Liu, H.T.; Liu, Z.P.; Xia, Q.F.; et al. Computational model of flood discharge splash in large hydropower stations. Journal of Hydraulic Research, 2015, 53(5), 576-587.]

 Point 17: Some key parameters are not mentioned. The rationale on the choice of the particular set of parameters should be explained with more details. Have the authors experimented with other sets of values? What are the sensitivities of these parameters on the results?

Response to reviewer: Thank you for your kindly suggestion. Actually, we have measured the temperature, humidity, PM2.0 and PM10.0 around the powerhouse, but these parameters are easily influenced by indoor environment, and the sensitivity of the above parameters to the shape of the flip bucket is poor. As the rainfall intensity distribution around the powerhouse is mainly subject to the splashing water weight and the nappe wind on the upstream slope of the powerhouse tailrace, Therefore, we used them as the main control indicators to study the rainfall intensity distribution of each bucket type. In this research, we mainly focus on the established water conservancy project, and we optimize the shape of the bucket with the principle of minimum construction and best effect.

 Point 18: Some assumptions are stated in various sections. Justifications should be provided on these assumptions. Evaluation on how they will affect the results should be made.

Response to reviewer: Thank you for your kindly suggestion,as you said, we have applied a lot of equations and assumptions in this paper, such as the splash flow q,the number n,the random distribution function of the ejection parameters(the diameter, initial velocity, initial elevation angle, the initial azimuth angle).As the mechanism of flood discharge atomization is extremely complex,

and it’s a tremendous difficult to use theoretical derivation method to obtain the mechanism. So we conducted a lot of splashing test and prototype data to obtain these semi-theoretical and semi-empirical formulas for calculating the flood discharge atomization, and it has been verified to be applicable to predict the atomization of a hydropower station by many researchers. Therefore, in this paper we still use these basic formulas.  [Lian, J.J.; Li, C.Y.; Liu, F. et al. A prediction method of flood discharge atomization for high dams. Journal of Hydraulic Research, 2014, 52(2), 274-282. Liu, H.T.; Liu, Z.P.; Xia, Q.F.; et al. Computational model of flood discharge splash in large hydropower stations. Journal of Hydraulic Research, 2015, 53(5), 576-587. Liu, F.; Huang, C.Y.; Yang, H. Comparative study of numerical result and field investigation for atomization of high dam. Journal of Hydroelectric Engineering, 2010, 29(1),19-23. (In Chinese). Liu, H.T.; Sun, S.K.; Wang, X.S. et al. Study on the distribution of splash intensity during nappe impingement. Journal of Hydrodynamics, 2009, 24(2), 217-223.(In Chinese)]

 Point 19: The discussion section in the present form is relatively weak and should be strengthened with more details and justifications.

Response to reviewer: Thank you for your kindly suggestion, in the revised manuscript,we have strengthened the discussion section as following:

l is the width of the impinging edge. The previous mathematical model of flood discharge atomization has significant disadvantages that the trajectory distance and the width of the impinging outer edge is calculated by the empirical formula[13.14,20,21]. The value of l are varies from one buckets to another. For the special shaped bucket (the tongue-shaped bucket and the twisted bucket ), the flow regime is extremely complicated, and the empirical formula is difficult to calculate the width and impact location of water jet impingement accurately, while the impinging width of water jet has a great influence on the rainfall intensity distribution. Therefore, we used the physical model test to obtain the specific hydraulic parameters of impinging outer edge of water jet which forms a linear ejection source and divide it into multiple segments.  [Lian, J.J.; Li, C.Y.; Liu, F. et al. A prediction method of flood discharge atomization for high dams. Journal of Hydraulic Research, 2014, 52(2), 274-282.;  Liu, F.; Huang, C.Y.; Yang, H. Comparative study of numerical result and field investigation for atomization of high dam. Journal of Hydroelectric Engineering, 2010, 29(1),19-23. (In Chinese); Zhang, H.; Lian J.J.; Li, H.P. Mathematical model of droplet randomly formed by splash of nappe. Journal of Hydraulic Engineering. 2003, 36(4): 430-433. (In Chinese); Lian, J.J., Liu, F., Zhang, H. Numerical simulation of atomization due to flood discharges of hydropower stations. Trans. Tianjin Univ. 2006, 12(5), 341–345.]

In the previous research of the stochastic mathematical model, the velocity and diameter of water droplets were taken as independent variable respectively[13,14,20,21,22]. However, through the model test, we find that there is a certain correlation between splashing droplet diameter and its velocity. With the droplet diameter increasing, the ejection required larger initial impulse. So for the large diameter droplets, the splash velocity and splash area is relatively small, and conversely the small diameter droplets have high velocity and wide splash area, and the relationship of velocity and diameter have a great influence on simulating of the rainfall intensity distribution. Therefore,in our research, we use a laser raindrop spectrograph to obtain the dimensionless relationship between the droplet diameter and splashing velocity. [Lian, J.J.; Li, C.Y.; Liu, F. et al. A prediction method of flood discharge atomization for high dams. Journal of Hydraulic Research, 2014, 52(2), 274-282.;  Liu, F.; Huang, C.Y.; Yang, H. Comparative study of numerical result and field investigation for atomization of high dam. Journal of Hydroelectric Engineering, 2010, 29(1),19-23. (In Chinese); Lian, J.J., Liu, F., Zhang, H. Numerical simulation of atomization due to flood discharges of hydropower stations. Trans. Tianjin Univ. 2006, 12(5), 341–345. Liu, H.T.; Sun, S.K.; Wang, X.S. et al. Study on the distribution of splash intensity during nappe impingement. Journal of Hydrodynamics, 2009, 24(2), 217-223.(In Chinese); Liu, H.T.; Liu, Z.P.; Xia, Q.F.; et al. Computational model of flood discharge splash in large hydropower stations. Journal of Hydraulic Research, 2015, 53(5), 576-587.]

Due to the mechanism of nappe wind is being extremely complex, there are few research on the distribution of nappe wind, and most previous stochastic mathematical model of flood discharge use generalization function to solve it[13,14,33]. However, the spatial distribution of nappe wind is vary from one bucket to another greatly, so the generalization function is unreasonable to deal with the downstream nappe wind. As the accuracy of the nappe wind has a great influence on the rainfall intensity distribution. Therefore, we use the ultrasonic wind sensor to measure the spatial distribution of the nappe wind. [Lian, J.J.; Li, C.Y.; Liu, F. et al. A prediction method of flood discharge atomization for high dams. Journal of Hydraulic Research, 2014, 52(2), 274-282.;  Liu, F.; Huang, C.Y.; Yang, H. Comparative study of numerical result and field investigation for atomization of high dam. Journal of Hydroelectric Engineering, 2010, 29(1),19-23. (In Chinese);  Liu, H.T.; Sun, S.K.; Wang, X.S. et al. Study on the distribution of splash intensity during nappe impingement. Journal of Hydrodynamics, 2009, 24(2), 217-223.(In Chinese)]

 Point 20: There are some occasional grammatical problems within the text. It may need the attention of someone fluent in English language to enhance the readability.

Response to reviewer: Thank you for your kindly suggestion, We have asked foreign experts to revise the grammar problem.

 Point 21:  Moreover, the manuscript could be substantially improved by relying and citing more on recent literatures about contemporary real-life case studies of optimization techniques in water resources engineering such as the followings:

  Fotovatikhah, F., et al., “Survey of Computational Intelligence as Basis to Big Flood Management: Challenges, research directions and Future Work,” Engineering Applications of Computational Fluid Mechanics 12 (1): 411-437 2018.

  Taormina, R., et al., “Neural network river forecasting through baseflow separation and binary-coded swarm optimization”, Journal of Hydrology 529 (3): 1788-1797 2015.

   Wu, C.L., et al., “Rainfall-Runoff Modeling Using Artificial Neural Network Coupled with Singular Spectrum Analysis”, Journal of Hydrology 399 (3-4): 394-409 2011.

 Wang, W.C., et al., “Improved annual rainfall-runoff forecasting using PSO-SVM model based on EEMD,” Journal of Hydroinformatics 15 (4): 1377-1390 2013.

 Cheng, C.T., et al., “Flood control management system for reservoirs,” Environmental Modeling & Software 19 (12): 1141-1150 2004.

 Chau, K.W., et al., “Use of Meta-Heuristic Techniques in Rainfall-Runoff Modelling” Water 9(3): article no. 186, 6p 2017.

Response to reviewer: Thank you for your kindly suggestion, as your said, we have citied more recent literatures about contemporary real-life case studies of optimization techniques in water resources engineering in the revised manuscript, including the “Survey of Computational Intelligence as Basis to Big Flood Management: Challenges, research directions and Future Work”, “Neural network river forecasting through baseflow separation and binary-coded swarm optimization”, “Rainfall-Runoff Modeling Using Artificial Neural Network Coupled with Singular Spectrum Analysis”, “Neural network river forecasting through baseflow separation and binary-coded swarm optimization”, “Improved annual rainfall-runoff forecasting using PSO-SVM model based on EEMD,”, “Flood control management system for reservoirs” and “Use of Meta-Heuristic Techniques in Rainfall-Runoff  Modeling” you have mentioned in the comments.

Point 22:   Some inconsistencies and minor errors that needed attention are:

l   Replace “…When the jet impact with…” with “…When the jet impacts with…” in line 203 of p.6

l   Replace “…of nappe wind is extremely complex…” with “…of nappe wind being extremely complex…” in line 242 of p.8

l   Replace “…fitting, As it shown in…” with “…fitting. As is shown in…” in line 281 of p.9

l   Replace “…Base on the random…” with “…Based on the random…” in line 321 of p.10

Response to reviewer: Thank you for your kindly suggestion, they has been corrected in the revised manuscript.

Round 2

Author Response

The Explanation

Manuscript ID: ijerph-410901. Title: “An improved empirical model for flood discharge atomization and its application to optimize the flip bucket of Nazixia project”.

To reviewer: 1 

Dear Editors and Reviewers:

Thanks for your kind letter and for the reviewers’ comments concerning our manuscript. Those comments are helpful for us to revise and improve our paper. We have studied comments carefully and tried our best to revise and improve the manuscript and made great changes in the manuscript according to the reviewers’ good comments. The responds to the reviewer’s comments are as following:

Comments to the Author 

OVERVIEW

I have appreciated the efforts by the Authors in replying to my concerns. The MS is improved, but it still shows significant drawbacks. Namely: (i) The results are qualitative, though they are of some interest,because scale effects are significant in my opinion; (ii) The manuscript is rather long without that being necessary (for example Table 1 is unneeded and relationships from 8 to 23 unessential because without general impact); (iii) More critical insights should be explicated in the manuscript also according to the comments in the “Authors’ response to Report 1”; (iv) the English language and style are poor and often hampering a proper understanding of the text. In addition, some terms are technically unsound.

(i)Response to reviewer: We are really appreciated for your suggestion. I agree with you that the scale effects exist in the physical model test about flood discharge atomization. Actually, we didn’t convert the rainfall intensity of the physical model to that of the prototype with the similarity law, but we compared the value of control indicators for atomization of the physical model test results to optimize the shape of flip bucket.We have added more description about the scale effects in the revised version as following:

The physical model test is an extension and supplementation to prototype observation, and as it is not subject to time and environment constraints, repetitive tests and quantitative descriptions can be carried out. However, the physical model test is usually designed according to the Froude similitude, and it would bring great uncertainty to the atomization model similarity, which is greatly affected by surface tension and air buoyancy [13]. Many researchers have studied the scale effects of the physical model. Wu et al. [12] proposed that when the Weber number is greater than 500, the influences of surface tension and viscous force are relatively weak and have little effect on the test results. While there is a close relationship between the Weber number and the model’s geometric dimensions, the similarity relations between the model and the prototype results should be explored further [13].

(ii) Response to reviewer: Thank you for your kindly suggestion, we have deleted a lot of unnessary expressions and repeated descriptions to shorten the manuscript, including deleting the Table 1 and relationships from 8 to 23 and other revisions as shown in the revised version.

(iii) Response to reviewer: Thank you for your kindly suggestion, and we have added more detailed descriptions in the revised version as the following:

Flip bucket types.

Generally, there are three types of flip bucket for trajectory energy dissipation:  the constant width-type;  the expansion-type; and  the contraction-type. In this research, we mainly focused on the established water conservancy project, and we optimized the shape of the bucket with the principles of minimum construction and best effect; therefore, we selected a typical flip bucket of each type to study their atomization characteristics, as shown in the Fig.2. Shape 1 is the original bucket of the hydropower station. At the same time, we found that the downstream scouring and silting of different buckets increased with the bucket angle decreased, and when the bucket angle was too small, this could seriously reduce the output of the power station. The splashing droplets and nappe wind near the powerhouse obviously increased with an increase in the bucket angle. When the bucket angle is too big, this can seriously aggravate the rainfall intensity. Therefore, we chose a bucket angle of 50o to study the rainfall intensity distribution of different flip buckets, and the test conditions are shown in Table 1.

The number of splashed water droplets.

When the jet flow impinges with the downstream water surface, taking the continuous bucket for example, a rectangle with a length  and width  is selected around any point on the outer boundary of the jet. The water volume q through the rectangle in unit time can be calculated as [13]

(1)

where , based on the feedback analysis of the prototype and experiment data,  , and the coefficient varies for different buckets. For the continuous bucket, the value of ranges from 0.0050.01, for the silt bucket and twisted bucket, the value of ranges from 0.010.03. h is the thickness of the jet flow, which is obtained by the physical model test,  is the projection of water droplet velocity in the Z-axis. Provided that q is completely converted to the splashed water droplets, the number (n) of splashed water droplets per unit time can be expressed as  . This has been verified in our previous research [13]; therefore, adopted it in this research. is the average diameter of water droplets, and a previous study [13] showed that the value of  is usually selected as a constant that equals 3 mm. C and CM are the water concentration coefficients at any point, and the maximum water concentration coefficient in the same cross-section, respectively . The relationship between C and CM can be expressed as [13, 33]:

(2)

(3)

where ; CM and are the maximum and average water concentration coefficients in the same cross-section, respectively. andare the distances from the measurement point to the jet centres in two curvilinear directions, b is the width of the jet flow, is the Froude number, which can be expressed as,  represents the thickness of the jet flow without aeration,  can be expressed as , and is the initial Froude number on the outlet of the bucket.

The impinging width of water jet l has a great influence on the rainfall intensity distribution. The previous mathematical model of flood discharge atomization calculated the trajectory distance and the width of the impinging outer edge by the empirical formula [13,14,20,21]. However, l varies from one bucket to another. For special shaped buckets (the tongue-shaped bucket and the twisted bucket), the flow regime is extremely complicated, and it is difficult to calculate the width and impact location of water jet impingement accurately with the empirical formula. So, we used the physical model test to obtain the specific hydraulic parameters of the impinging outer edge of the water jet which forms a linear ejection source, and we divided it into multiple segments.

Stochastic model of splash water droplets

Due to the large velocity of the trajectory nappe, the outer edge of the jet is severely aerated and discontinuous and it cannot completely enter the downstream water surface [15], and most droplets splash around, which provides the main source of atomization rainfall [34]. The mechanism of flood discharge atomization is extremely complex, and it is tremendously difficult to use the theoretical derivation method to obtain the mechanism. So, we conducted many splashing tests and gathered prototype data to obtain the semi-empirical formula, and it has been verified to be applicable for predicting the atomization by many researchers [13,14,20,21]. In addition, Liu et al. [34] and Liu et al. [35] compared the calculated results of the stochastic splash model with the experimental data, and their results indicated that the gamma distribution assumption of the diameter and initial velocity can simulate the atomization phenomena very well; therefore, we adopted this approach in this research. 

 (iv) Response to reviewer: Thank you for your kindly suggestion, we have asked the professional English editing service of MDPI for extensive English editing.

SPECIFIC COMMENTS

As general recommendation, in case of Figures and Tables the Authors should specify when they refer to the physical model or to the prototype! There is some confusion all over the manuscript.

Response to reviewer: Thank you very much for your well-intentioned suggestion. We have checked and specified the Figures and Tables in the revised version as the following:

Figure 1. Arrangement of the physical model test

Figure 2. Flip bucket types of the physical model test: (a) shape 1;(b) shape 2;(c) shape 3; (d) shape 4; (units: cm)

Table 1. Prototypical discharge conditions

Figure 3. The dimensionless relationship between the droplet diameter and velocity of the physical model test.

Figure.4 Nappe wind distribution in the longitudinal direction in the physical model test

Figure.5 Nappe wind distribution in the vertical direction in the physical model test

Figure.6 Nappe wind distribution in the lateral direction in the physical model test: a shape 1 and shape 3;bshape 2 and shape 4.

Figure.7 Prototype observation of powerhouse rainfall

Figure.8 Rainfall intensity distribution of prototypical numerical results (unit: mm h−1)

Figure.9 Rainfall intensity distribution of different bucket types of the prototype: (a) rainfall intensity  distribution of shape 1 and shape 2; (b) rainfall intensity distribution of shape 3 and shape 4 (unit: mm h−1).

Table 2. The weight of splash water for different bucket types in the physical model test

Figure.10 Splash weight at the bottom of the left slope  in the physical model test

Figure.11 Nappe wind at the upstream slope in the physical model test

Figure.12 Rainfall intensity distribution of original and recommended buckets with different prototype conditions: acondition 1; bcondition 2; ccondition 4; dcondition 5; econdition 7 (units: mm h−1)

 Table 3. The splash water weight under different conditions in the physical model test

Figure.13 Splash weight at the bottom of the left slope of the physical model test: a represents the splash weight of condition 1 to condition 2; brepresents the splash weight of condition 3 to condition 7

Figure.14 Nappe wind at the upstream slope of the powerhouse of the physical model test:arepresents the nappe wind of condition1 to condition 3;brepresents the nappe wind of condition4 to condition 7. 

 [TITLE] It reads “mathematical”, but I believe “empirical” would be more proper.

Response to reviewer: Thank you for your kindly suggestion, we agree with your advise, in the revised version, the “mathematical”has been replaced by the“empirical”.

[ABSTRACT] (i) Is the part of the text from line 13 (where it reads “The previous”) to line 18 (where it reads “is unreasonable”) appropriate for an abstract? I guess not. On the other hand literature shortcomings are well (and properly) explicated in the section Introduction; (ii) Line 22. It reads “the mathematical mode”, but it should read “mathematical model”; (iii) Line 29. It reads “rangs from”, but it should read “ranges from”; (iv) Line 31. Where it reads “And it shows that”, I would write “Finally, it is shown that”.

Response to reviewer: Thank you very much for your well-intentioned suggestion. We have  revised these problem as the following:

(i) We agree with your advice, in the revised version, we have revised the abstract section as the following:

Abstract: Flood discharge atomization is a serious challenge that threatens the daily lives of the residents around the dam area as well as the safety of water conservancy project. This research aims to improve the prediction accuracy of the stochastic splash model. A physical model test with four types of flip bucket is conducted to obtain the hydraulic parameters of the impinging outer edge of the water jet, the relationship of the splashing droplet diameter with its corresponding velocity, and the spatial distribution of the downstream nappe wind. The factors mentioned above are introduced to formulate the empirical model. The rule obtained from the numerical analyses is compared with the results of the physical model test and the prototype observations, which yields a solid agreement. The numerical results indicate that the powerhouse is no longer in the heavy rain area when adopting the flip bucket whose curved surface is attached to the left wall. The rainfall intensity of the powerhouse is significantly weaker than that of other types under the designed condition, so we choose it as the recommended bucket type. Meanwhile, we compare the rainfall intensity distribution of the original bucket and the recommended bucket under different discharge which rates ranging from 150.71 to 1094.9 m3/s. It is found that the powerhouse and the owner camp are no longer in the heavy rain area under all of the working conditions. Finally, it is shown that the atomization influence during the flood discharge can be reduced by using the recommended bucket.

(ii) Thanks for pointing out the mistakes in the manuscript, it has been corrected in the revised manuscript.

(iii) Thanks for pointing out the mistakes in the manuscript, it has been corrected in the revised manuscript.

(iv)Thanks for your kindly suggestion, it has been corrected in the revised manuscript.

[INTRODUCTION] (i) Line 37. A space should be inserted between “valleys.” And “When”; (ii) Line 42. It reads “on[4-6]..”, but it should read “on [4-6].”; (iii) Line 73. It reads “Liu [15-18] has studied”, but I would write“Liu et al. [15-18] have studied”, according to the References section; (iv) Line 76. It reads “Lian and Zhang[19-21]”, but I would write “Zhang et al. [19,20] and Lian et al. [21]”, according to the References section; (v) Line 79. It reads “Based the research of Lian and Zhang, Liu[14] have developed”, but it should read “Based on the researches of Lian and Zhang, Liu et al. [14] have developed”; (vi) Line 82. It reads “Liu[22]”, but it should read “Liu et al. [22]”, according to the References section; (vii) Line 87. It reads “Liu[31]”, but it should read “Liu et al. [31]”, according to the References section.

Response to reviewer: Thank you very much for your well-intentioned suggestion. We have  revised these problem as the following:

(i) Thanks for pointing out the mistakes in the manuscript, it has been corrected in the revised manuscript.

(ii) Thanks for pointing out the mistakes in the manuscript, “on[4-6]..” has been replaced by “on [4-6].”in the revised manuscript.

(iii) Thanks for pointing out the mistakes in the manuscript, “Liu [15-18] has studied” has been replaced by “Liu et al. [15-18] have studied”in the revised manuscript.

(iv) Thanks for pointing out the mistakes in the manuscript, “on[4-6]..” has been replaced by “on [4-6].”in the revised manuscript.

(iv)Thanks for pointing out the mistakes in the manuscript, “Lian and Zhang[19-21]”has been replaced by “Zhang et al. [19,20] and Lian et al. [21]”in the revised manuscript.

(v) Thanks for pointing out the mistakes in the manuscript, “Based the research of Lian and Zhang, Liu[14] have developed” has been replaced by“Based on the researches of Lian and Zhang, Liu et al. [14] have developed”in the revised manuscript.

(vi) Thanks for pointing out the mistakes in the manuscript, “Liu[22]”has been replaced by“Liu et al. [22]”in the revised manuscript.

(vii) Thanks for pointing out the mistakes in the manuscript, “Liu[31]”has been replaced by“Liu etal. [31]”in the revised manuscript.

[PHYSICAL MODEL] (i) In this section a discussion on the scale effects should be explicated. The Authors made this discussion in the “Authors’ response to Report 1” document, but it would be suitable to bring those comments also in the manuscript; (ii) Lines 126 and 127. Is “discharge structures” technically correct? May be it would be more proper using “overflow structures”, “outlet structures”, “diversion structures”… (iii) Is Table 1 really needed? All the values can be inferred from the model geometric scale (1:50) and the Froude similarity! (iv) Figure 1. It reads “Tailrace channal”, but it should read “Tailrace channel”! (v) Figure 1. It

reads “device of spash water”, but it should read “device of splash water”! (vi) Figure 1. It reads “Represent the nappe”, but it should read “Represents the nappe”; (vii) Line 134. Is “flood pressure” technically correct? (viii) Line 148. It reads “types iss compared”, but it should read “type is compared”; (ix) Line 156. Where it reads “Isbash formula”, I would add a citation [..] to a suitable paper; (x) Lines from 156 to 158. It remains unclear how the sediment particle size for the mobile bed of the physical model was determined. If I have understood correctly, also when the Isbash formula is considered, the scouring velocity should be known…..How did the Authors determine the scouring velocity? (xi) Figure 2 caption. The unit of measurement should be specified! (xii) Table 2. I would specify that upstream and downstream levels are given in meters above sea level! (xiii) Table 2. What does “Gate opening  5 m” stand for? It is clear that in this case gates are partially open, but “5 m” should be related to the maximum opening height…..

Response to reviewer: Thank you very much for your well-intentioned suggestion. We have  revised these problem as the following:

(i) We agree with your advice, and we have added more comments about the scale effects in the revised version as following:

The physical model test is an extension and supplementation to prototype observation, and as it is not subject to time and environment constraints, repetitive tests and quantitative descriptions can be carried out. However, the physical model test is usually designed according to the Froude similitude, and it would bring great uncertainty to the atomization model similarity, which is greatly affected by surface tension and air buoyancy [13]. Many researchers have studied the scale effects of the physical model. Wu et al. [12]  proposed that when the Weber number is greater than 500, the influences of surface tension and viscous force are relatively weak and have little effect on the test results. While there is a close relationship between the Weber number and the model’s geometric dimensions, the similarity relations between the model and the prototype results should be explored further [13].

(ii) Thanks for pointing out the mistakes in the manuscript, “discharge structures” has been replaced by “outlet structures”in the revised manuscript.

(iii) We agree with your advice, and the“Table 1” has been deleted in the revised manuscript.

(vi) Thanks for pointing out the mistakes in the manuscript, it has been corrected in the revised manuscript.

(v) Thanks for pointing out the mistakes in the manuscript, it has been corrected in the revised manuscript.

(vi) Thanks for pointing out the mistakes in the manuscript, it has been corrected in the revised manuscript.

Figure 1. Arrangement of the physical model test

 (vii) Thanks for pointing out the mistakes in the manuscript, it has been corrected in the revised manuscript.

(viii) Thanks for pointing out the mistakes in the manuscript, it has been corrected in the revised manuscript.

(ix) We agree with your advice, we have added a citation in the revised manuscript. And we added more description in the revised version as the following:

Based on the scouring velocity data supported by the Geological Survey Institute, we use the Isbash formula[Ministry of Water Resources of the PRC, SL155-2012, spacification for normal hydrualic model test,[S],Beijing, China Water Power Press, 2012.]

(x) Thanks for your kindly suggestion, the scouring velocity data is supported by the Gansu Institute of Water Conservancy and Hydropower.  

(xi) Thanks for pointing out the mistakes in the manuscript, and the unit of measurement have be specified in the revised manuscript.

(xii) Thanks for pointing out the mistakes in the manuscript, “upstream and downstream levels”has been replaced by“upstream and downstream altitude”in the revised manuscript.

(xiii) Thanks for your kindly suggestion, the “Gate opening height” has been replaced by the “Gate opening ratio” as the following:

Table 1. Prototypical discharge conditions

Operating condition

Height above sea levelm

Gate opening ratio

(m)

Discharge

(m3/s)

Upstream

Downstream

1 (Check water level)

3202.38

3090.00

full open

1094.90

2 (Design water level)

3201.70

3090.00

full open

1006.00

3 (Normal water level)

3201.50

3090.00

full open

972.00

4

3201.50

3090.00

50.00%

848.53

5

3201.50

3090.00

25.00%

531.70

6

3201.50

3090.00

10.00%

350.62

7

3201.50

3090.00

5.00%

150.71

 [BASIC THEORY OF THE IMPROVED STOCHASTIC MODEL] (i) In my previous review I remarked the following issues: Flip bucket types. With reference to Figure 2, the scheme in Figure2a represents the original bucket of the hydropower station while the remainders represent alternative buckets. Could the Authors justify/discuss the ratio behind the shapes of the alternative buckets? The number of splashed water droplets. Could the Authors explain/discuss the equations they provide for the total splash flow q and the number n of splashed water droplets? Overall, this section requires more explanation behind equations and coefficients provided in. Stochastic model of splash water droplets. Could the Authors explain/discuss physical reasons according to the diameter d of water droplet and the initial velocity v0 of water droplet would obey to a Gamma distribution (instead, for instance, to a normal distribution)? The Authors provide suitable comments in the “Authors’ response to Report 1” document, but it would be suitable to bring those comments also in the manuscript; (ii) Line 197. It reads “the jet flow impinge with”, but it should read “the jet flow impinges with”; (iii) Line 201. It reads “the coefficient is varies from”, but it should read “the coefficient varies from”; (iv) Line 207. It reads “the value of l are varies from”, but it should read “the value of l varies from”; (v) Line 217. It reads “is the maximum and the average”, but it should read “are the maximum and the average”; (vi) Line 235. It reads “observtion”, but it should read “observation”; (vii) Line 237. It reads “the randomness”, but it should read “The randomness”; (viii) Line 250. It reads “it is shown in Fig. 3,”, but it should read “as shown in Fig. 3.”; (ix) Line 268. It reads “is vary from”, but I would write “is different from”; (x) Figure 4 caption. What does “in axis directions” stand for? (x) Line 362. It reads “are used”, but it should read “is used

Response to reviewer: Thank you very much for your well-intentioned suggestion. We have  revised these problem as the following:

(i)We agree with your advice, and we have added more detailed descriptions in the revised version as the following:

Flip bucket types.

Generally, there are three types of flip bucket for trajectory energy dissipation:  the constant width-type;  the expansion-type; and  the contraction-type. In this research, we mainly focused on the established water conservancy project, and we optimized the shape of the bucket with the principles of minimum construction and best effect; therefore, we selected a typical flip bucket of each type to study their atomization characteristics, as shown in the Fig.2. Shape 1 is the original bucket of the hydropower station. At the same time, we found that the downstream scouring and silting of different buckets increased with the bucket angle decreased, and when the bucket angle was too small, this could seriously reduce the output of the power station. The splashing droplets and nappe wind near the powerhouse obviously increased with an increase in the bucket angle. When the bucket angle is too big, this can seriously aggravate the rainfall intensity. Therefore, we chose a bucket angle of 50o to study the rainfall intensity distribution of different flip buckets, and the test conditions are shown in Table 1.

The number of splashed water droplets.

When the jet flow impinges with the downstream water surface, taking the continuous bucket for example, a rectangle with a length  and width  is selected around any point on the outer boundary of the jet. The water volume q through the rectangle in unit time can be calculated as [13]

(1)

where , based on the feedback analysis of the prototype and experiment data,  , and the coefficient varies for different buckets. For the continuous bucket, the value of ranges from 0.0050.01, for the silt bucket and twisted bucket, the value of ranges from 0.010.03. h is the thickness of the jet flow, which is obtained by the physical model test,  is the projection of water droplet velocity in the Z-axis. Provided that q is completely converted to the splashed water droplets, the number (n) of splashed water droplets per unit time can be expressed as  . This has been verified in our previous research [13]; therefore, adopted it in this research. is the average diameter of water droplets, and a previous study [13] showed that the value of  is usually selected as a constant that equals 3 mm. C and CM are the water concentration coefficients at any point, and the maximum water concentration coefficient in the same cross-section, respectively . The relationship between C and CM can be expressed as [13, 33]:

(2)

(3)

where ; CM and are the maximum and average water concentration coefficients in the same cross-section, respectively. andare the distances from the measurement point to the jet centres in two curvilinear directions, b is the width of the jet flow, is the Froude number, which can be expressed as,  represents the thickness of the jet flow without aeration,  can be expressed as , and is the initial Froude number on the outlet of the bucket.

The impinging width of water jet l has a great influence on the rainfall intensity distribution. The previous mathematical model of flood discharge atomization calculated the trajectory distance and the width of the impinging outer edge by the empirical formula [13,14,20,21]. However, l varies from one bucket to another. For special shaped buckets (the tongue-shaped bucket and the twisted bucket), the flow regime is extremely complicated, and it is difficult to calculate the width and impact location of water jet impingement accurately with the empirical formula. So, we used the physical model test to obtain the specific hydraulic parameters of the impinging outer edge of the water jet which forms a linear ejection source, and we divided it into multiple segments.

Stochastic model of splash water droplets

Due to the large velocity of the trajectory nappe, the outer edge of the jet is severely aerated and discontinuous and it cannot completely enter the downstream water surface [15], and most droplets splash around, which provides the main source of atomization rainfall [34]. The mechanism of flood discharge atomization is extremely complex, and it is tremendously difficult to use the theoretical derivation method to obtain the mechanism. So, we conducted many splashing tests and gathered prototype data to obtain the semi-empirical formula, and it has been verified to be applicable for predicting the atomization by many researchers [13,14,20,21]. In addition, Liu et al. [34] and Liu et al. [35] compared the calculated results of the stochastic splash model with the experimental data, and their results indicated that the gamma distribution assumption of the diameter and initial velocity can simulate the atomization phenomena very well; therefore, we adopted this approach in this research. 

(ii)Thanks for pointing out the mistakes in the manuscript, it has been corrected in the revised manuscript.

(iii)Thanks for pointing out the mistakes in the manuscript, it has been corrected in the revised manuscript.

(iv)Thanks for pointing out the mistakes in the manuscript, it has been corrected in the revised manuscript.

(v)Thanks for pointing out the mistakes in the manuscript, it has been corrected in the revised manuscript.

(vi)Thanks for pointing out the mistakes in the manuscript, it has been corrected in the revised manuscript.

(vii)Thanks for pointing out the mistakes in the manuscript, it has been corrected in the revised manuscript.

(viii)Thanks for pointing out the mistakes in the manuscript, it has been corrected in the revised manuscript.

(ix)Thanks for pointing out the mistakes in the manuscript, it has been corrected in the revised manuscript.

(x)Thanks for your kindly suggestion, the“in axis directions”has been replaced by the “in  longitudinal direction”, and the longitudinal direction stands for the direction of the spillway axis.

(xi)Thanks for pointing out the mistakes in the manuscript , it has been corrected in the revised manuscript.

[VERIFICATION OF THE IMPROVED STOCHASTIC MODEL] (i) Lines 372 and 373. It reads “Therefore, The numerical results correspond well to the prototype observations”, but how did the Authors ascertain this comparison if field data are missing? (ii) Figure 8 caption. This caption appears incomplete! (iii) Line 399. It reads “the plash collecting device”, but it should read “the splash collecting device”; (iv) Line 432. It reads “is obvious smaller”, but I would write “is obviously smaller”; (v) Line 443. It reads “tend to decrease”, but it should read “tends to decrease”; (vi) Lines from 468 to 476. Note how this sentence is 9 text-lines long! It is unreadable!

Response to reviewer: Thank you very much for your well-intentioned suggestion. We have  revised these problem as the following:

(i)According to the field data of the rainfall intensity supported by Qinghai Yellow River Medium-sized Hydropower Development Company, the data indicated that the the powerhouse is still in the rainstorm area and partly in the torrential rainstorm area, which is consistent with the numerical results.We have described more detail in the revised Ms as the following:

The field data of the rainfall intensity supported by the administration of Nazixia hydropower station indicated that the powerhouse is partly in the torrential rainstorm area, and a photograph of the powerhouse rainfall is shown in Fig.7. The numerical results are shown in the Fig.8, the powerhouse is still in the rainstorm area and partly in the torrential rainstorm area. Therefore, the numerical results correspond well to the prototype observations.

(ii)Thanks for your kindly suggestion, we changed the Figure 8 caption as the following :

Figure.8 Rainfall intensity distribution of prototypical numerical results (unit: mm h−1)

(iii)Thanks for pointing out the mistakes in the manuscript , it has been corrected in the revised manuscript.

(iv)Thanks for pointing out the mistakes in the manuscript, , it has been corrected in the revised manuscript.

(v)Thanks for pointing out the mistakes in the manuscript, it has been corrected in the revised manuscript.

(vi) We are sorry for that the English problem, we have revised it as the following:

To compare the rainfall intensity distribution of the recommended bucket with that of the original bucket under different working conditions, the optimized mathematical model was used to calculate the rainfall intensity distribution, and the numerical results are shown in Fig.12. The intensity and coverage area of the rainfall reduced as the discharge decreased. The rainfall coverage area of the recommended bucket shifted to the right side of the spillway axis. This is mainly because the impact point and the peak value of the nappe wind turned to the right side of the spillway axis. The rainfall intensity around the powerhouse is significantly reduced when we adopted the recommended bucket, and the powerhouse was no longer in the rainstorm area under different working conditions.

Reviewer 2 Report

The revised paper has addressed all my previous comments, and I suggest to ACCEPT the paper as it is now.

Author Response

The Explanation

Manuscript ID: ijerph-410901. Title: “An improved empirical model for flood discharge atomization and its application to optimize the flip bucket of Nazixia project”.

To reviewer: 2 

Comments to the Author 

The revised paper has addressed all my previous comments, and I suggest to ACCEPT the paper as it is now.

Thanks for your kind letter and for the reviewers comments concerning our manuscript. Your comments are helpful for us to revise and improve our paper. We have asked the professional English editing service of MDPI for extensive English editing in this round.

Round 3

Author Response

Thanks for your kind letter and for the reviewers’ comments concerning our manuscript. Those comments are helpful for us to revise and improve our paper. We have studied comments carefully and tried our best to revise and improve the manuscript and made great changes in the manuscript according to the reviewers’ good comments. The responds to the reviewer’s comments are as following

Comments to the Author 

(i)   Lines 74 and 75. It reads on the basis its, but it should read on the basis of its;

Response to reviewer: Thanks for pointing out the mistake in the manuscript, in the revised manuscript, The on the basis itshave been replaced by theon the basis of its.

(ii)  Lines 78 and 80. The acronyms RB and RBF should be spelled out;

Response to reviewer: Thanks for pointing out the mistake in the manuscript, in the revised manuscript, The  RB and RBFhave been replaced by theback propagationand radial basis function.

(iii)            Line 176. It reads therefore, adopted, but it should read therefore, we adopted;

Response to reviewer: Thanks for pointing out the mistake in the manuscript, in the revised manuscript, The therefore, adoptedhave been replaced by thetherefore, we adopted;.

(iv) Line 206. It reads Liu et al. [34], but it should read Duan et al. [34] according to the References section;

Response to reviewer: Thanks for pointing out the mistake in the manuscript, in the revised manuscript, The Liu et al. [34]have been replaced by theLiu et al. [34].

(v)  Line 245. It reads shown in Fig.4, We defined, but it should read shown in Fig.4. We defined;

Response to reviewer: Thanks for pointing out the mistake in the manuscript, in the revised manuscript, Theshown in Fig.4, We definedhave been replaced by theshown in Fig.4. We defined.

(vi) Lines 287 and 288. Where it reads x i or x i  , it should read x i ’’ according to Equation (8). Analogously, where it reads u i ’’ , it should read u i ;

Response to reviewer: Thanks for pointing out the mistake in the manuscript, we have checked and modified all the formulas in the revised manuscript.

(vii)          Table 3. It reads Recommend bucket(g), but I would write Recommended bucket (g);

Response to reviewer: Thanks for pointing out the mistake in the manuscript, in the revised manuscript, TheRecommend bucket(g),have been replaced by theRecommended bucket (g).

(viii)         Line 417. It reads increasd, but it should read increased;

Response to reviewer: Thanks for pointing out the mistake in the manuscript, in the revised manuscript, Theincreasd,have been replaced by theincreased.

(ix) Line 442. It reads In this research, we presented an improved mathematical model., but I would specify as follows In this research, we presented an improved mathematical model for flood discharge atomization.;

Response to reviewer: Thanks for pointing out the mistake in the manuscript, in the revised manuscript, TheIn this research, we presented an improved mathematical model.,have been replaced by theIn this research, we presented an improved mathematical model for flood discharge atomization..

(x)  Line 453. It reads with that of and the original, but it should read with that of the original;

Response to reviewer: Thanks for pointing out the mistake in the manuscript, in the revised manuscript, Thewith that of and the original,have been replaced by thewith that of the original.

(xi) Ref.32. Please check this reference carefully. As examples, it reads spacification, but it should read specification, and it reads hydrualic, but it should read hydraulic!

Response to reviewer: Thanks for pointing out the mistake in the manuscript, in the revised manuscript, The last sentence of the abstract spacificationand hydrualichave been replaced by thespecificationand hydraulic, respectively. And we have checked the full manuscript carefully.

Thank you again for concerning our manuscript, and wish everything goes well with your work sincerely.